# Non-catalytic role of phosphoinositide 3-kinase in mesenchymal cell migration through non-canonical induction of p85β/AP2-mediated endocytosis

Hideaki T. Matsubayashi [1,2,3] ✉, Jack Mountain[1,2], Nozomi Takahashi[3], Abhijit Deb Roy[1,2], Tony Yao [1,2], Amy F. Peterson[1,2], Cristian Saez Gonzalez [1,2], Ibuki Kawamata [4,5,6] & Takanari Inoue [1,2] ✉

Class IA phosphoinositide 3-kinase (PI3K) galvanizes fundamental cellular processes such as migration, proliferation, and differentiation. To enable these multifaceted roles, the catalytic subunit p110 utilizes the multi-domain, regulatory subunit p85 through its inter SH2 domain (iSH2). In cell migration, its product PI(3,4,5)$P_3$ generates locomotive activity. While non-catalytic roles are also implicated, underlying mechanisms and their relationship to PI(3,4,5)$P_3$ signaling remain elusive. Here, we report that a disordered region of iSH2 contains AP2 binding motifs which can trigger clathrin and dynamin-mediated endocytosis independent of PI3K catalytic activity. The AP2 binding motif mutants of p85 aberrantly accumulate at focal adhesions and increase both velocity and persistency in fibroblast migration. We thus propose the dual functionality of PI3K in the control of cell motility, catalytic and non-catalytic, arising distinctly from juxtaposed regions within iSH2.

Class IA PI3Ks are lipid kinases that catalyze phosphatidylinositol (3,4,5)-triphosphate (PI(3,4,5)$P_3$) production[1,2]. In the canonical growth factor pathway, PI(3,4,5)$P_3$ production leads to Akt/mTOR activation and subsequent upregulation of proliferation and survival. Besides this primary function, PI3K and PI(3,4,5)$P_3$ manifest versatile roles in many other physiological contexts including vesicular trafficking, differentiation, immune reaction, and cell migration[2–5]. Due to its multi-tasking roles, the PI3K catalytic function is modulated by various interaction partners such as ubiquitin ligase Cbl-b[6], tumor suppressor BRD7[7], thyroid hormone receptor β[8], transmembrane tyrosine phosphatase CD148[9], and microtubule-associated protein MAP4[10].

Class IA PI3K is a heterodimeric complex composed of a catalytic subunit (p110α, p110β, or p110δ) and a regulatory subunit (p85α, p55α, p50α, p85β, or p55γ)[1,11,12]. Upon activation of receptor tyrosine kinases (RTKs), such as platelet-derived growth factor (PDGF) receptors in fibroblasts, nSH2 and cSH2 domains in the regulatory subunit recognize tyrosine phosphorylation on the receptors and adapter molecules[13,14]. As regulatory subunits tightly associate with p110 through an inter SH2 domain (iSH2) that resides between two SH2 domains[11], p110 consequently accumulates at the plasma membrane. The phosphotyrosine binding of SH2 domains liberates their inhibitory contact with p110[15,16], thus resulting in signal-specific PI3K activation proximal to its substrate, phosphoinositide (4,5)-bisphosphate.

The catalytic activity of PI3K is one of the major positive regulators in cell migration. In amoeboid cells such as *Dictyostelium*

---

[1]Department of Cell Biology, School of Medicine, Johns Hopkins University, Baltimore, MD, USA. [2]Center for Cell Dynamics, Institute of Basic Biomedical Sciences, Johns Hopkins University, Baltimore, MD, USA. [3]Frontier Research Institute for Interdisciplinary Sciences, Tohoku University, Tohoku, Japan. [4]Department of Robotics, Tohoku University, Tohoku, Japan. [5]Natural Science Division, Ochanomizu University, Kyoto, Japan. [6]Present address: Graduate School of Science, Kyoto University, Kyoto, Japan. ✉e-mail: hideaki.matsubayashi.e1@tohoku.ac.jp; jctinoue@jhmi.edu

*discoideum* and mammalian neutrophils, chemoattractant induces PI(3,4,5)P₃ accumulation at the front of cells[17–19], leading to activation of the Rho family of small GTPases including Rac1[19–21] and cell protrusions driven by the actin cytoskeleton. Mesenchymal cells such as fibroblasts also establish similar PI(3,4,5)P₃ polarity[22]. However, a recent study found that PI3K in fibroblasts acts as an amplifier of nascent lamellipodia instead of an initiator of protrusion[23]. Further research found that this PI3K-actin feedback loop originates from nascent adhesions, another unique feature of mesenchymal cell migration[24]. Therefore, amoeboid and mesenchymal cells utilize distinct mechanisms, at least at the level of PI3K, with heretofore elusive mechanisms.

In the face of the catalytic-role-centric studies, non-catalytic roles of p85 have also been reported. In ER stress response, p85 brings XBP-1s to the nucleus to upregulate unfolded protein response genes[25,26]. p85 is also involved in receptor internalization through the interaction with an adapter molecule insulin receptor substrate 1 (IRS-1), Rab GTPases activation, or ubiquitination on p85 itself[27–29]. In addition, p85 regulates cytoskeletal reorganization in concert with the small GTPase Cdc42[30,31]. It therefore is important to consider PI3K as a multifaceted molecule to fully understand its functions and regulations.

In this study, we combine bioinformatics and chemical biology approaches with live-cell fluorescence imaging to reveal a non-catalytic function of PI3K in which a part of the p85β iSH2 domain induces endocytosis mediated by clathrin and dynamin. Using p85 knockout cells with genetic rescues, we show that this non-catalytic induction of endocytosis regulates cell migration properties through local regulation of p85 at focal adhesions.

## Results

### The iSH2 domain of regulatory subunit p85 has AP2 binding motifs

To explore possible non-catalytic roles of PI3K, we analyzed the primary sequence of the regulatory subunits of class IA PI3K (p85α, p85β, and p55γ). Using Eukaryotic Linear Motif (ELM) prediction[32], we found that the iSH2 domain of the C-terminal region of p85β accommodates three consensus binding motifs for AP2[33], an adapter protein for clathrin-mediated endocytosis, namely YxxΦ, di-leucine, and acidic clusters (Fig. 1a, Supplementary Fig. 1a). Consistent with the crystal structure of p110 complexed with iSH2-cSH2[16], primary sequence analysis from IUPred2A[34], PrDOS[35], and PONDR[36] also predicted the C-terminal region of iSH2 to be intrinsically disordered and unlikely to be a part of secondary structures (Supplementary Fig. 1a). We then used AlphaFold2 (AF2) to simulate the interaction between the AP2 complex and the predicted AP2 binding motifs. Taking amino acid sequences of four subunits of the AP2 core from a crystal structure (PDB: 2JKR)[37] and 16 amino acids including three potential AP2 binding motifs from mouse p85β (ETEDQYSLMEDEDALP), we found AF2 predicted that all five structures show iSH2 binding to either the μ subunit (rank 1, 2, 3, 4) or the σ subunit (rank 5). The binding configurations resemble the previously characterized crystal structures of AP2-YxxΦ complex[38] and AP2-di-Leucine complex[37], respectively (Fig. 1b, Supplementary Fig. 1b–h). Furthermore, all-atom molecular dynamics (MD) simulations corroborated the computational plausibility of these interactions. (Supplementary Fig. 1i, j). These results suggest possible interaction between p85 and AP-2, which could lead to endocytosis upon their membrane targeting.

We then tested direct interaction between iSH2 and AP2 by pull-down assay using purified AP2 core and GST-fused iSH2. While wild type iSH2 pulled down AP2 effectively, the amount of pulled down AP2 was reduced to less than half by an L601A mutation in the Di-leucine motif and almost completely abolished by replacing the motif region with 3×SAGG flexible linker (motifGS) (Fig. 1c. Supplementary Fig. 2). These data show that AP2 binding motifs in iSH2 directly and specifically interact with AP2 complex.

### Plasma membrane recruitment of the iSH2 domain induces endocytosis

Whether a given molecule is capable of inducing endocytosis can be tested by recruiting such molecules to plasma membranes[39,40]. With the help of a chemically inducible dimerization (CID) system[41], we aimed to recruit iSH2 including the putative AP2 binding motifs to the plasma membrane and see if this results in endocytosis. To this end, we used rapamycin-dependent heterodimerization of FK506-binding protein (FKBP) and FK506-rapamycin-binding domain (FRB) to trap YFP-FKBP-iSH2 (YF-iSH2) at plasma membrane-anchored Lyn-CFP-FRB (Lyn-CR). Within several minutes after accumulation of YF-iSH2 at the plasma membrane, numerous mobile puncta became visible in the cytosol (Fig. 1d, Supplementary Movies 1–3). The puncta were seen only with YF-iSH2 but not with a negative control YFP-FKBP (YF), suggesting that iSH2 is responsible for induction of puncta derived from the plasma membrane.

We then tested colocalization between the observed puncta and markers for endocytosis. When we used a membrane staining dye mCLING[42], which is internalized to endomembranes upon endocytosis, the puncta colocalized well with the dye (Supplementary Fig. 3). Furthermore, the iSH2 puncta also colocalized with other markers such as mCherry-Rab5 (early endosome) and Lamp1-mRFP (lysosome), but not with negative controls such as mCherry (cytosol) and mCherry-KDEL (ER) (Fig. 1e).

Endocytic activity is highly sensitive to ambient temperature, likely due to critical involvement of dynamin GTPase which has an unusually high $Q_{10}$ temperature coefficient value[43,44]. When conducting iSH2 recruitment to the plasma membrane at a reduced temperature (37 °C to 23 °C), we observed much fewer puncta (Supplementary Fig. 4a–c, Supplementary movies 1–3). This is consistent with the lack of documentation of such puncta upon iSH2 recruitment by our group and others in the past[45–48]. Collectively, these results strongly support the idea that membrane-recruited iSH2 induces endocytosis.

### iSH2-mediated endocytosis is context independent

To test how well the iSH2-mediated endocytosis can be generalized, we repeated the CID recruitment assay with two modifications. First, we used FRB anchored to the plasma membrane through six different targeting sequences (Supplementary Data 1). In all cases except KRas4B-CAAX, we observed puncta formation (Supplementary Fig. 4d, e). Furthermore, the endocytosis can be also triggered by a light inducible dimerization system (iLID-SspB)[49] (Supplementary Fig. 4f). Thus, iSH2-mediated endocytosis is not specific to a certain type of plasma membrane targeting or dimerization scheme.

### iSH2-mediated endocytosis depends on the AP2 binding motifs

To determine if the predicted AP2 binding motifs are necessary for iSH2-mediated endocytosis, we deleted 12 amino acids (aa) within the motif clusters (Δmotif) or replaced them with a 3×SAGG linker (motifGS). When the recruitment assay was conducted with each of these iSH2 mutants, we saw little to no puncta, indicating the necessity of the 12 aa for inducing endocytosis (Fig. 1f, Supplementary Fig. 5). Then, we individually mutated the YxxΦ motif, di-leucine motif, and acidic cluster. Whereas point mutations in the di-leucine motifs drastically decreased endocytic activity, Y to A mutation in the YxxΦ motif did not show significant effect (Fig. 1f, Supplementary Fig. 5). Replacement of the acidic cluster EDEDA with GSAGG partially reduced the endocytic activity (Fig. 1f, Supplementary Fig. 5). These results suggest that the di-leucine motif and acidic clusters contribute to iSH2-mediated endocytosis.

### iSH2-mediated endocytosis depends on clathrin and dynamin

To gain insights into the cellular mechanisms of iSH2-mediated endocytosis, we first examined *in cellulo* interaction between iSH2

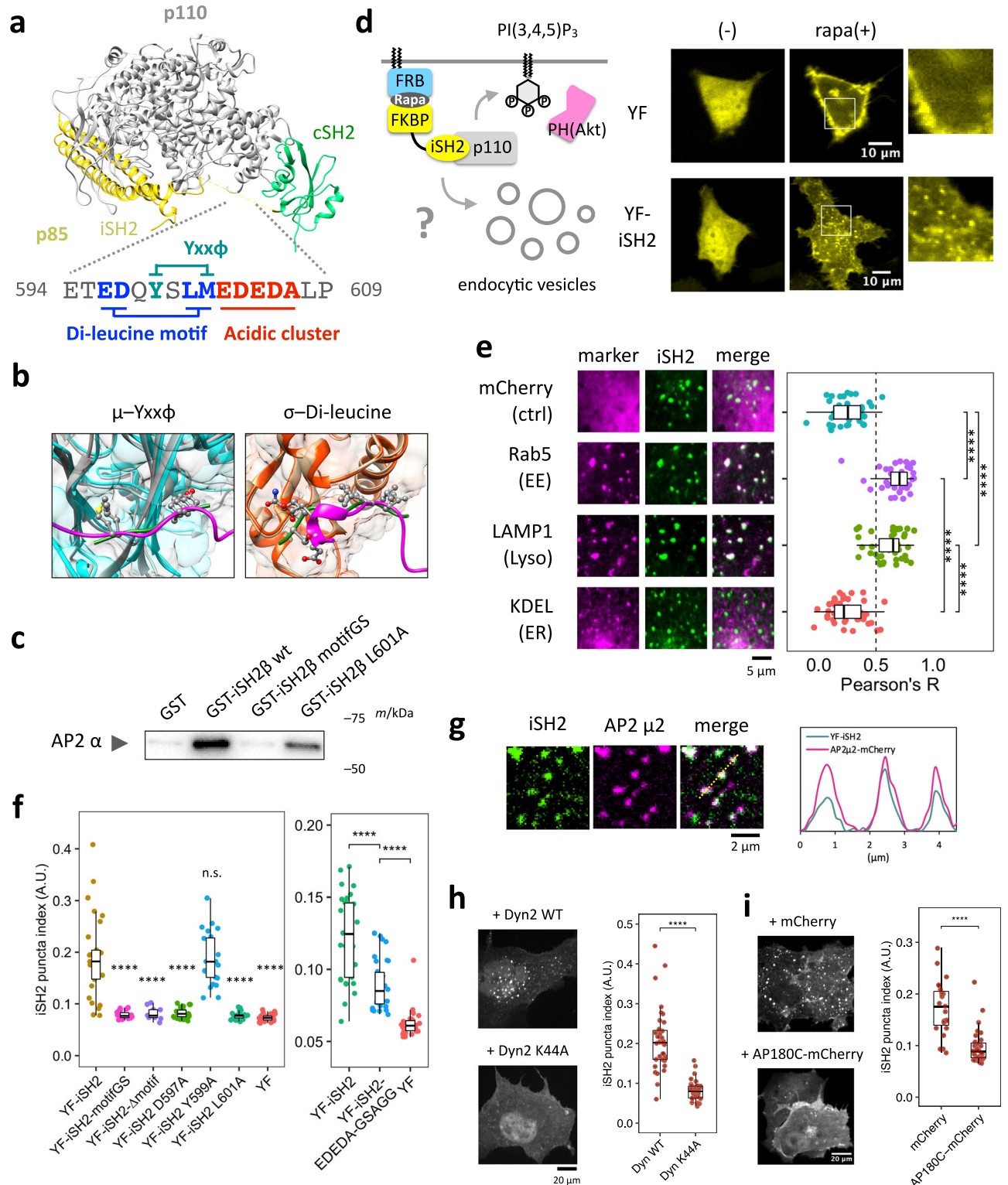

and AP2 by applying an inducible co-recruitment assay[50,51] (Supplementary Fig. 6a). Here, we recruit an iSH2 domain to the plasma membrane using the chemically inducible dimerization scheme, and measure how much of a bait protein, AP2, gets co-recruited under TIRF microscopy. After recruitment of YFP-FKBP-labeled iSH2 to the plasma membrane, we observed an increase in the fluorescence intensity of AP2-mCherry (co-recruitment index, CI: 1.23), but not mCherry control construct (CI: 1.03) (Supplementary Fig. 6b, c), verifying that iSH2 and AP2 interact with each other in cells. This AP2 co-recruitment was reduced when we used iSH2 motif mutants,

Δmotif (CI: 1.07) and motifGS (CI: 1.20) (Supplementary Fig. 6b, c). Similarly, we measured the extent of colocalization between AP2 and iSH2 after recruitment of iSH2 to the plasma membrane. AP2 fluorescence signals on the plasma membrane colocalized with the membrane-recruited iSH2, but not with the motif mutant (Fig. 1g, Supplementary Fig. 6d, e). These results suggest that the AP2 binding motif of p85 binds to and colocalizes with AP2 on the plasma membrane.

Of note, colocalization of iSH2 and AP2 was observed when FRB-CFP-CAAX(KRas4B) was used as a plasma membrane anchor

**Fig. 1 | Plasma membrane recruitment of the iSH2 domain induces clathrin and dynamin dependent endocytosis. a** Crystal structure of PI3K (PDB 2Y3A) and the AP2 binding motifs of mouse p85β iSH2 domain. **b** AF2-predicted structures of the 16 aa iSH2 peptide binding to AP2. Left: rank 2 structure showing the iSH2 YxxΦ motif binding to μ subunit. Right: rank 5 structure showing the iSH2 di-Leucine motif binding to the σ subunit. Gray: AF2-predicted μ subunit, Orange: AF2-predicted σ subunit, Magenta: AF2-predicted 16 aa iSH2 peptide, Cyan and Green (left): μ subunit and TGN38 peptide from PDB 2XA7. Tan and Green (right): σ subunit and CD4 peptide from PDB 2JKR. Motif amino acids are represented in ball-stick model. **c** Western blot of the AP2 α subunit for pulldown assay using GST-iSH2 and AP2 core. Similar results were obtained from 3 independent experiments. **d** Confocal images of endocytic vesicles produced by plasma membrane targeting of the iSH2 domain. HeLa cells were transiently transfected with Lyn-ECFP-FRB, mCherry-PH(Akt), and EYFP-FKBP or EYFP-FKBP-iSH2. Images show before and after 100 nM rapamycin addition. Similar results were obtained from more than 3 independent experiments. **e** Confocal images of iSH2-induced vesicles colocalized with endocytosis marker molecules: mCherry-Rab5 (early endosome) and LAMP1-mRFP (lysosome). mCherry (cytosol) and mCherry-KDEL (ER) were used as negative controls. The graph shows Pearson's correlation between iSH2 and marker molecules. mCherry, n = 40 cells. Rab5, n = 39 cells. LAMP1, n = 41 cells. KDEL, n = 41 cells. P values: mCherry−Rab5, $<1.0 \times 10^{-12}$. mCherry−LAMP1, $<1.0 \times 10^{-12}$. Rab5−KDEL, $<1.0 \times 10^{-12}$. LAMP−KEDL, $2.4 \times 10^{-12}$. **f** Quantified iSH2-mediated endocytosis indices (see Methods) of wild type iSH2 and mutants in AP2 binding motifs. YF-iSH2, n = 29 cells. motifGS, n = 24 cells. Δmotif, n = 12 cells. D597A, n = 24 cells. Y599A, n = 24 cells. L601A, n = 21 cells. YF, n = 31 cells. YF-iSH2, n = 32 cells. EDEDA-GSAGG, n = 33 cells. YF, n = 31 cells. P values: (left) motifGS, $1.1 \times 10^{-7}$. Δmotif, $1.0 \times 10^{-4}$. D597A, $5.4 \times 10^{-7}$. Y599A, 1.0. L601A, $3.1 \times 10^{-7}$. YF, $1.7 \times 10^{-9}$. (right) YFiSH2−EDEDA-GSAGG, $9.5 \times 10^{-6}$. EDEDA-GSAGG−YF, $3.3 \times 10^{-10}$. **g** TIRF images of iSH2 vesicles colocalized with AP2. Similar results were obtained from more than 3 independent experiments. **h, i** Confocal images of iSH2 vesicles showing dynamin and clathrin dependency. Vesicle formation was suppressed in the presence of the dominant negative form of dynamin (K44A) or AP180C. Dyn WT, n = 37 cells. Dyn K44A, n = 37 cells. mCherry, n = 30 cells. AP180C-mCherry, n = 40 cells. P values: (**h**) $5.3 \times 10^{-12}$, (**i**) $1.2 \times 10^{-7}$. Box whisker plots represent median, 1st, 3rd quartiles and 1.5×inter-quartile range. P values: *: <0.05, **: <0.01, ***: <0.001, ****: <0.0001. n.s.: not significant. **e, f** Steel-Dwass test (two sided). In the left panel of (**f**), only p values against YF-iSH2 are shown. **h, i** Wilcoxon rank sum test (two sided).

(Supplementary Fig. 6d, e), despite the poor endocytosis induction of CAAX(KRas4B) (Supplementary Fig. 4d, e). This result suggest that while iSH2 interacts with AP2 regardless of the type of plasma membrane anchor, vesicle maturation and/or membrane remodeling are somehow stalled with KRas4B-CAAX anchor. We then tested two dominant negative mutants, N-terminus truncated AP180 (AP180C)[52,53] and GTPase-defective dynamin (Dyn2-K44A)[54,55], that inhibit endocytic processes. These mutants significantly reduced the numbers of endocytosed puncta, suggesting that iSH2-mediated endocytosis depends on clathrin and dynamin (Fig. 1h, i). Taken together, we conclude that iSH2 brings AP2 to the plasma membrane, which triggers endocytosis through clathrin and dynamin.

## iSH2-mediated endocytosis is independent of PI3K catalytic activity

Catalytic activity of PI3K and its product PI(3,4,5)P₃ have been implicated in various types of endocytosis[56–59]. Since the iSH2 domain binds to endogenous p110 and its plasma membrane recruitment leads to PI(3,4,5)P₃ production[45–48], we asked if iSH2-mediated endocytosis is dependent on PI(3,4,5)P₃. We tested this with either a PI3K inhibitor (LY294002) or a deletion mutant of iSH2 (iSH2-DN). LY294002 binds to the ATP binding pocket of p110 and inhibit its catalytic function[60], whereas iSH2-DN mutation abolishes iSH2-p110 interaction[61] and decouples iSH2 from PI(3,4,5)P₃-Akt activation (Supplementary Fig. 7a, b). When we performed the iSH2 recruitment assay in the presence of either of these reagents, puncta formation occurred normally despite the production of PI(3,4,5)P₃ being suppressed in the same cells (Fig. 2a, Supplementary Fig. 7c). This indicates that iSH2-mediated endocytosis is independent of the p110 kinase activity and can be classified as a non-catalytic function of PI3K.

We further tested the role of the p110 catalytic subunit in iSH2-mediated endocytosis. Expression of kinase-dead p110α (D915N) did not affect iSH2-mediated endocytosis compared to the mCherry control (Supplementary Fig. 7d, e), consistent with a dispensable role of the PI3K catalytic activity in iSH2-mediated endocytosis. In contrast, expression of wild type p110α suppressed iSH2-mediated endocytosis, which appeared to be dependent on the expression level of p110 (Supplementary Fig. 7d, e). While the mechanism by which wild type p110 suppresses iSH2-mediated endocytosis remains unknown, excessive PI(3,4,5)P₃ production due to overexpressed p110 may sequester AP2 and/or clathrin molecules from iSH2. Nevertheless, these data support the notion that iSH2-mediated endocytosis is independent of p110 kinase activity.

## iSH2-mediated endocytosis is β isoform specific

The iSH2 domain is defined in all three regulatory subunits of class IA PI3K (p85α, p85β, and p55γ)[1]. We took iSH2 domains from different isoforms of human and mouse and asked if iSH2-mediated endocytosis is conserved among them by using the CID recruitment assay. iSH2 from p85β (both human and mouse) induced endocytosis, but α or γ isoforms did not (Fig. 2b, Supplementary Fig. 7f), indicating that endocytic activity is β isoform specific. Of interest, mouse iSH2α pulled down AP2 at a level comparable to mouse iSH2β in vitro (Supplementary Fig. 2). Thus, although the exact mechanism is unknown, kinetic parameters that are not captured in pulldown assay, such as dissociation rate, may be involved in the isoform specificity.

## The 46 aa disordered region is necessary and sufficient for iSH2-mediated endocytosis

The iSH2 domain has been considered as a single domain whose main role is to bind to p110 and bring the catalytic subunit to the plasma membrane upon receptor stimulation. To locate exactly which part of iSH2 contributes to p110 binding, and which part contributes to the endocytosis induction, we performed a sequential truncation to the iSH2 domain. As a result, the C-terminal 46 aa was found to be both necessary and sufficient to induce endocytosis (Fig. 2c, Supplementary Fig. 7g–k). Consistently, the 46 aa region was sufficient to directly interact with AP2 in pulldown assays (Supplementary Fig. 2). In contrast, PI(3,4,5)P₃ production remained intact with iSH2 lacking this 46 aa region (Fig. 2c, Supplementary Fig. 7j, k). Our results demonstrate that the iSH2 domain can be structurally and functionally separated into two regions - the p110 binding coiled-coil region for catalytic actions and the 46 aa disordered region encoding AP2 binding motif for non-catalytic induction of endocytosis.

## Generation of MEF cell lines with p85β AP2 binding motif mutants and their biochemical characterization

To investigate how the unexpected link between p85β and AP2 influences the cellular functions of PI3K, we took advantage of p85α/β double knock out (DKO) in mouse embryonic fibroblasts (MEFs)[62] to which a series of p85 variants were individually introduced via lentiviral infection (Supplementary Fig. 8a). DKO allowed us to study mutations in AP2 binding motifs of p85β as well as isoform differences between p85α and p85β in a p85 knockout background. Since both the di-leucine motif and the acidic cluster contribute to endocytic activity (Fig. 1f), we created two p85β mutants whose 12 aa motif region was either truncated or replaced with 3×SAGG, serving as AP2 motif deficient forms of p85β. YFP was tagged on the rescued p85 to sort the

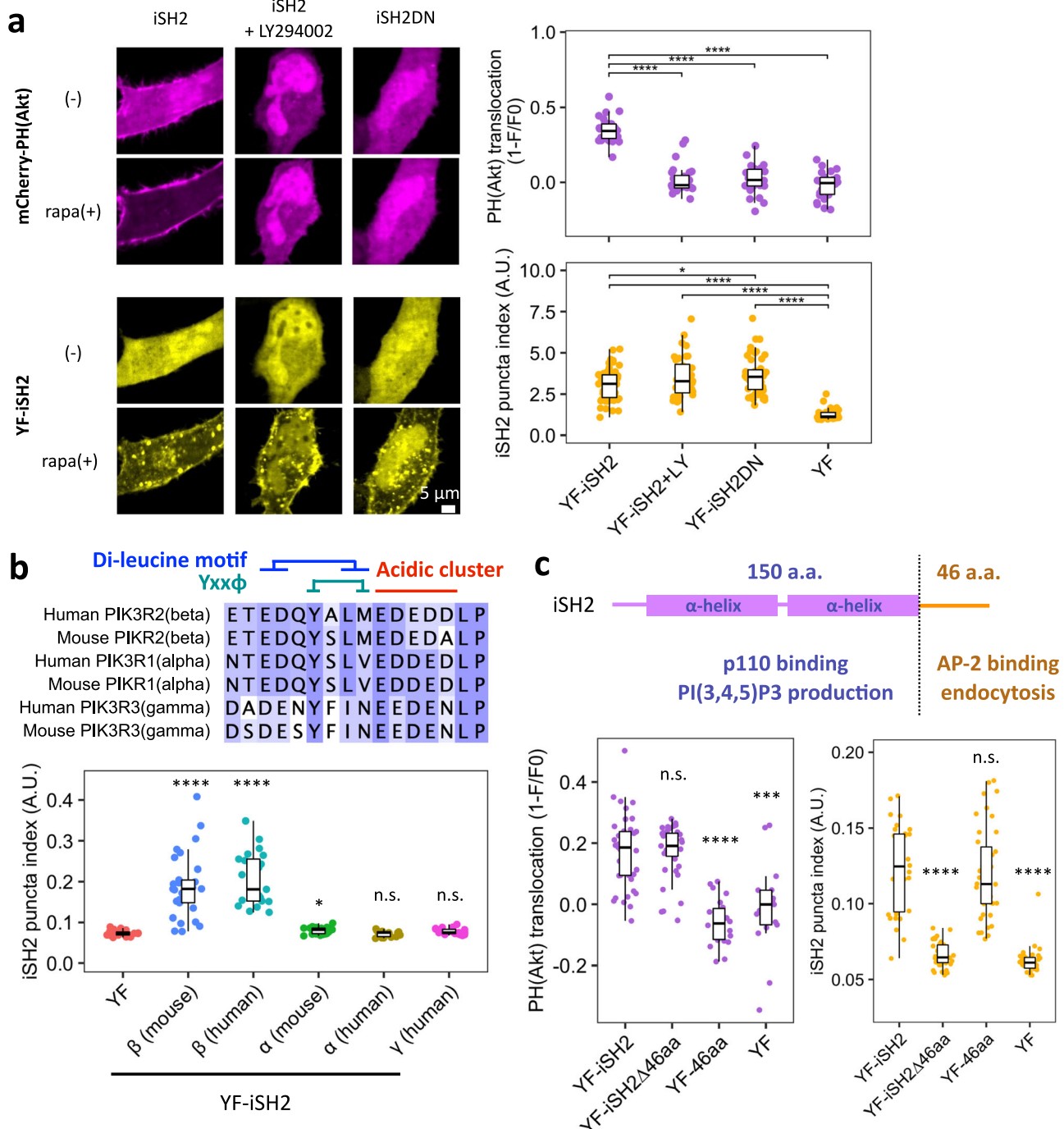

**Fig. 2 | iSH2-mediated endocytosis is independent of PI3K catalytic activity and the C-terminal 46 aa region is necessary and sufficient. a** Confocal images of PI(3,4,5)P3 sensor PH(Akt) and iSH2 vesicles. Quantifications are shown on the right. LY294002: PI3K inhibitor (50 μM), iSH2(DN): deletion mutant lacking p110 binding site. YF-iSH2, n = 59 cells. YF-iSH2+LY, n = 36 cells. YF-iSH2DN, n = 47 cells. YF, n = 43 cells. P value of PH(Akt) translocation: YF-iSH2−YFiSH2-LY, $8.6 \times 10^{-10}$. YF-iSH2−YFiSH2DN, $7.0 \times 10^{-10}$. YF-iSH2−YF, $8.8 \times 10^{-10}$. P value of iSH2 puncta index: YF-iSH2−YFiSH2DN, 0.049. YF-iSH2−YF, $<1.0 \times 10^{-12}$. YF-iSH2+LY−YF, $<1.0 \times 10^{-12}$. YF-iSH2DN−YF, $<1.0 \times 10^{-12}$. **b** Top: Amino acid sequence alignment of the AP2 binding motif region of human and mouse p85α, p85β, p55γ isoforms. Bottom: Quantification of iSH2 vesicles produced by each isoform. YF, n = 31 cells. β (mouse), n = 29

cells. β (human), n = 21 cells. α (mouse), n = 23 cells. α (human), n = 17 cells. γ (mouse), n = 19 cells. P values: β (mouse), $1.2 \times 10^{-9}$. β (human), $1.9 \times 10^{-8}$. α (mouse), 0.029. α (human), 0.99. γ (mouse)=0.29. **c** Secondary structure of the mouse p85β iSH2 domain and quantification of PH(Akt) translocation and iSH2 vesicles. YF-iSH2, n = 32 cells. YF-iSH2Δ46aa, n = 38 cells. YF-46aa, n = 39 cells. YF, n = 31 cells. P value of PH(Akt) translocation: YF-iSH2Δ46aa, 0.99. YF-46aa, $1.6 \times 10^{-8}$. YF, $2.0 \times 10^{-4}$. P value of iSH2 puncta index: YF-iSH2Δ46aa, $4.4 \times 10^{-11}$. YF-46aa, 0.96. YF, $3.1 \times 10^{-10}$. Box whisker plots represent median, 1st, 3rd quartiles and 1.5×inter-quartile range. P values: *: <0.05, **: <0.01, ***: <0.001, ****: <0.0001. n.s.: not significant. Only p values against YF are shown in (b). Only p values against YF-iSH2 are shown in (c). **a−c** Steel-Dwass test (two sided).

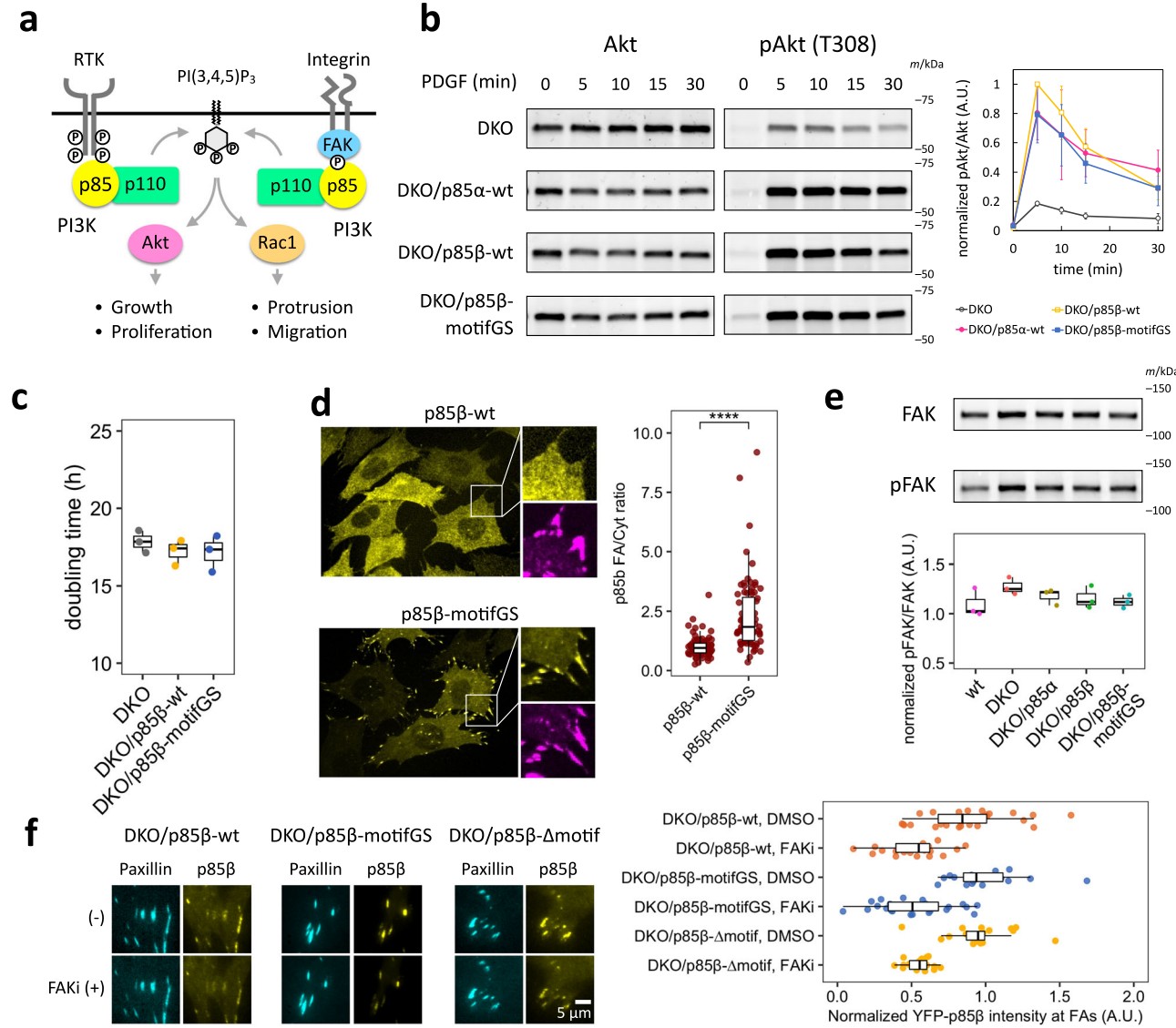

**Fig. 3 | Mutation in AP2 binding motifs of p85β increases focal adhesion localization. a** Schematic of receptor tyrosine kinase-dependent and focal adhesion-dependent PI3K pathways. **b** Western blot of total- and phospho-Akt (T308) and its quantification. Cells were treated with 50 ng/mL PDGF for indicated time. pAkt/Akt level was normalized to DKO/p85β-wt 5 min. Data are presented as mean ± standard deviations. **c** Doubling time of DKO and p85 rescued MEF cells. **d** Confocal images of p85β-wt and p85β-motifGS cells and their quantification. Yellow: EYFP-p85β, Magenta: immunofluorescence against vinculin. p85β-wt, n = 70 cells. p85β-motifGS, n = 80 cells. P value: 6.8×10⁻¹⁵. **e** Western blot of total- and phospho-FAK (Y397) and its quantification. **f** FAK activity dependency of p85 focal adhesion localization. Cells were treated with DMSO or 10 μM PF-573228 (FAK inhibitor; FAKi) for 5 min and EYFP-p85β intensity were divided by the values of time=0. p85β-wt DMSO, n = 27 cells. p85β-wt FAKi, n = 20 cells. p85β-motifGS DMSO, n = 16 cells. p85β-motifGS FAKi, n = 23 cells. p85β-Δmotif, n = 20 cells. p85β-Δmotif FAKi, n = 18 cells. **b**, **c**, **e** n = 3 independent experiments. Box whisker plots represent median, 1st, 3rd quartiles and 1.5×inter-quartile range. P value: ****: <0.0001. **d** Wilcoxon rank sum test (two sided).

virus-infected cells and validate the consistency in the expression level of rescued p85 variants (Supplementary Fig. 8b).

Using these genetic resources, we first assessed a possible regulatory role of the AP2 binding motif in a receptor tyrosine kinase (RTK) pathway (Fig. 3a). Consistent with a previous report[62], expression of wild type p85β in DKO MEFs could rescue the elevated levels of Akt phosphorylation (pTyr-308) in response to PDGF addition (Fig. 3b). However, when we tested this with the mutant p85β cell lines, there was no significant difference from the wild type p85β. In assessing cell proliferation, we then found similar proliferation rates for cells rescued with wild type and motifGS mutant p85β (Fig. 3c). Thus, mutations in the AP2 binding motif of p85β did not show an apparent effect on Akt response or cell growth. Considering the possibility that AP2 binding of p85β regulates receptor internalization, we next

measured the effect on ERK, the other major pathway regulated by endocytic traffic of RTK[63]. The result showed similar dynamics in ERK response between wild type and mutant rescued cells (Supplementary Fig. 8c). We also tested the effect on transferrin receptors, a typical cargo of clathrin-dynamin endocytosis, and found no significant change in transferrin internalization between wild type and mutant rescue cells (Supplementary Fig. 8d). Therefore, the binding between p85β and AP2 did not seem to influence RTK signaling or general endocytic functions.

**Mutations in the AP2 binding motifs cause localization of p85β at focal adhesions**

Besides the RTK response, PI3K locally controls cellular morphodynamics in association with focal adhesions[24,30,64,65]. To determine if the

AP2 binding motifs are involved in such subcellular regulation, we next investigated the intracellular localization of wild type and mutant p85β using confocal microscopy. The motifGS p85 cell lines showed significantly enhanced accumulation at focal adhesions (Fig. 3d). Previous studies found that p85 localizes to focal adhesions through the interaction between the SH3 domain of p85 and auto-phosphorylated tyrosine (pY397) of focal adhesion kinase (FAK)[64,66–69]. We thus tested the effect of the AP2 motif mutation on FAK. Western blot analysis did not detect significant differences in the expression or phosphorylation level of FAK among the p85-rescued cell lines (Fig. 3e, Supplementary Fig. 9a). Using TIRF microscopy, we further performed live-cell imaging of p85 fused to YFP which was co-expressed with a focal adhesion marker mCerulean3-Paxillin[70] in the presence or absence of an FAK inhibitor PF-573228[71]. The results showed that both wild type and mutant p85 dissociated from focal adhesions after FAK inhibition with identical kinetics (Fig. 3f, Supplementary Fig. 9b–e). Together, the data suggest that the AP2 binding motifs are involved in sequestration of p85β from focal adhesions. Since the observed sequestration did not affect the interaction between the SH3 domain of p85β and pY397 of FAK, there is another mechanism underlying triggering of the sequestration.

### Fibroblasts with impaired AP2 binding motifs migrate faster and more persistently

Focal adhesions function as a molecular clutch for a cell to transmit mechanical force to the external environment[72], while simultaneously serving as a biochemical hub for PI3K-Rho GTPase-actin to extend lamellipodial protrusion[24,65]. Since mutations in the AP2 binding motifs altered localization of p85β at focal adhesions, we hypothesized that the AP2 binding motifs regulate cell migration through focal adhesions. To test this, we characterized migratory properties in a series of DKO MEFs in the presence of 10% FBS to trigger random migration (Fig. 4a, Supplementary Fig. 10a). DKO MEFs exhibited slower migration speed than wild type counterpart MEFs (Fig. 4b), consistent with the reduced Rac activity and less lamellipodia formation in the knockout cells[62]. Interestingly, rescuing the DKO cell line with wild type p85β further decreased migration speed (Fig. 4b, c). In contrast, the cells rescued with p85α or AP2 binding motif mutants of p85β did not show the decrement, suggesting that the AP2 motif negatively regulates migration (Fig. 4b, c, Supplementary Fig. 10a–c).

Both the dominant negative mutation of p85 (DN), which lacks the 470 to 504 aa residues necessary for p110 binding and thereby decouples receptors from PI3K signaling[61], and pharmacological inhibition of PI3K and FAK completely suppressed the migration. This basal level of migration was significantly lower than the migration activity of wild type p85β-rescued cells (Fig. 4b, Supplementary Fig. 10a–c). These results suggest that p85β has two layers of regulations on cell migration: positive regulation through PI3K catalytic product PI(3,4,5)P$_3$ and negative regulation through AP2-mediated sequestration of p85β from focal adhesions.

We then calculated the persistence ratio of cell motility defined as the ratio between displacement (d) and the total path length (D), which decreased over the course of migration assays. The decrease in wild type p85β-rescued cells was more prominent over time than mutant p85-rescued cells, suggesting that the link between p85 and AP2 is involved in a negative regulation of cell migration with a temporal delay from PI(3,4,5)P$_3$-mediated positive regulation (Fig. 4d, Supplementary Fig. 10c).

Difference in migration speed between wild-type p85 rescue cells and AP2 motif mutant rescue cells was also seen with PDGF as a stimulant, instead of FBS (Supplementary Fig. 10d), suggesting that the AP2-mediated motility control is at play under growth factor signaling. When measured by PH(Akt) translocation to the plasma membrane, wild-type p85β-rescued and motifGS p85β-rescued cells showed similar PI(3,4,5)P$_3$ response to PDGF stimulation within 30 min

(Supplementary Fig. 10e). Thus, PI3K signaling difference in longer time scale or subcellular activity may account for p85β-AP2-mediated negative regulation of cell migration.

### The role of the AP2 binding motif in chemotaxis

To test migration behavior in a physiologically relevant context, we performed chemotaxis assays where cells are guided to migrate in a directed manner according to a chemoattractant gradient (Fig. 4e). In line with the random migration results, p85β-rescued cells migrated more slowly than DKO, p85α-rescued, and p85β motif mutant-rescued cells (Fig. 4f, g). Although the persistence ratio yielded slightly different curves from those of random migration, wild type p85β-rescued cells consistently showed the least persistency among the tested cells (Fig. 4h). These data support the negative regulation of chemotaxis by AP2-mediated endocytosis. To examine its role in gradient sensing during chemotaxis, we quantified the forward migration index (FMI) defined as the ratio between forward displacement (y) and the total path length (D) (Fig. 4i). As a result, there was no significant difference in FMI among the conditions tested; wild type cells, DKO cells, and DKO cells rescued with p85α, p85β, or p85β-motifGS (Fig. 4j). These data suggest that AP2-mediated endocytosis downregulates migration properties such as speed and persistence, but not gradient sensing, during chemotaxis.

## Discussion

The iSH2 domain is characterized as a positive regulator of PI3K since the iSH2 domain is necessary to engage the catalytic activity of p110 and link the catalytic subunit to receptor-binding p85[61,73,74]. Our present study demonstrates that the iSH2 domain of p85β has concurrent negative regulation of cell migration through AP2-mediated endocytosis which originates from the C-terminal disordered region. Disruption of this linkage between p85β and AP2 led to abnormal accumulation of p85β at focal adhesions (Fig. 3d) and also increased speed and persistency of cell migration (Fig. 4). Based on these findings, we propose that the iSH2 domain, originally assigned as a single domain for a single function, consists of two parts with distinct, antagonistic functions: the p110 binding coiled-coil region to promote cell migration, and the AP2 motif-encoding disordered region to induce endocytosis for negative regulation of cell migration. One may wonder why PI3K elicits two opposing signals for cell motility control. Such a seemingly meaningless regulation may be explained by the kinetic difference. Upon stimulation, PI(3,4,5)P$_3$ production can initiate within milliseconds to seconds timescale[75], while clathrin-mediated endocytosis occurs more gradually (tens of seconds to a few minutes)[76]. The temporal difference creates an autonomous delayed negative feedback loop, which is one of the signature characteristics necessary for self-organized signal transduction often proposed in directed cell migration[77]. Thus, for PI3K to send out counteracting signals of different kinetics may be of importance for this intricate cell function.

We also determined that the AP2 motif regulates p85β localization at focal adhesions. Since cell protrusion signaling consisting of PI3K and actin is closely coupled with cell adhesions[23,24,65], sequestration of PI3K from focal adhesions could act as a negative regulator of chemotaxis. Considering that mutations to the AP2 binding motif did not affect the expression level or FAK phosphorylation (Fig. 3e), the p85-mediated endocytosis likely regulates the signals downstream of PI3K without drastically altering molecular composition of the focal adhesions. Interestingly, under PDGF stimulation, mutations in the AP2 binding motif increased cell migration speed without affecting other major pathway effectors such as Akt and ERK (Fig. 3b, c, Supplementary Fig. 8c, 10d). How does AP2-mediated regulation discriminate a specific signaling molecule from others? Two interesting observations may be of help to answer this question -the AP2 binding motif resides within the intrinsically disordered region (Supplementary Fig. 1), and

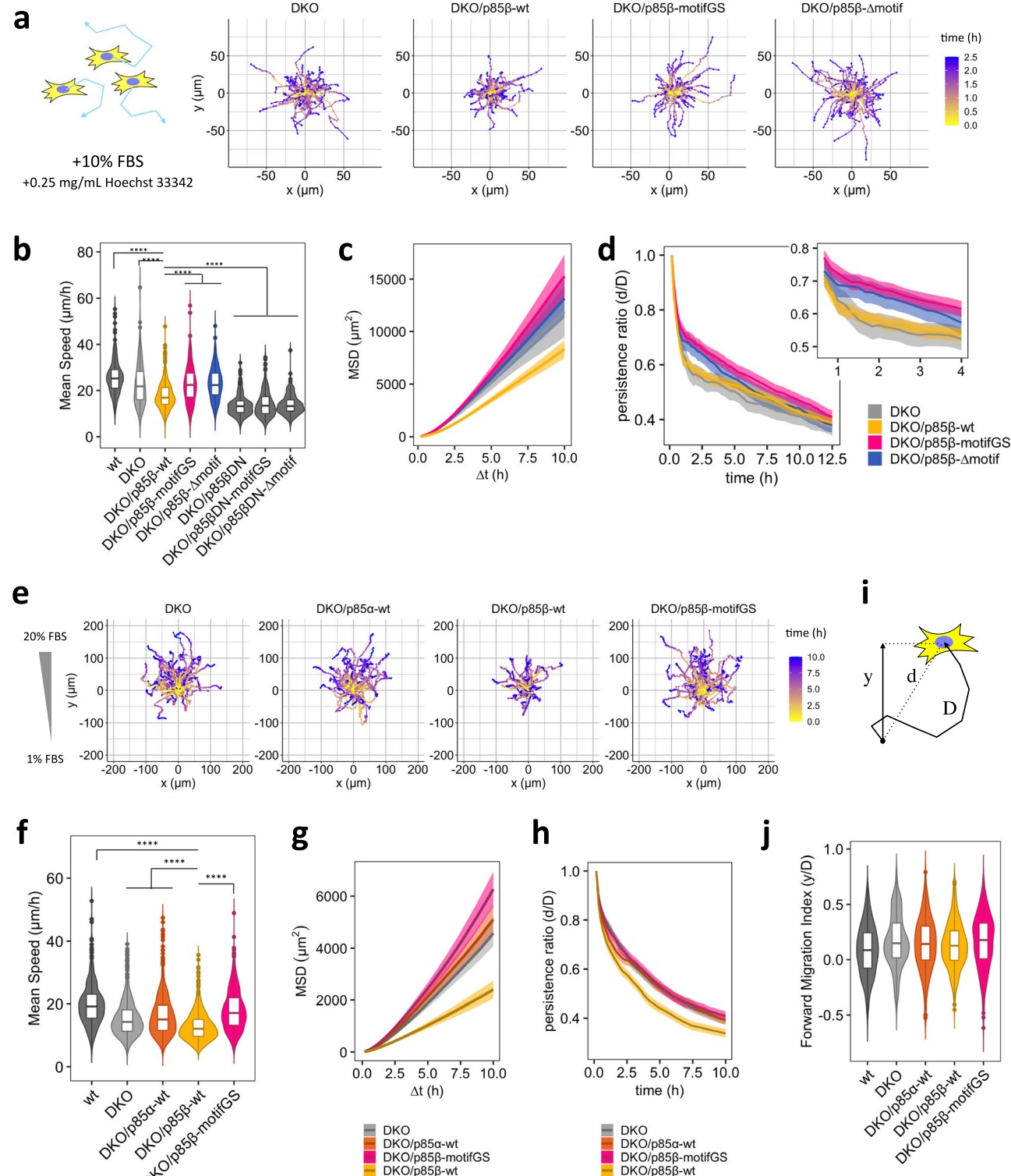

**Fig. 4 | Mutation in AP2 binding motifs of p85β enhances cell motility in random and chemotactic migration. a** Representative tracks of 2D random migration on fibronectin coated plates. Cells were allowed to migrate at 37 °C with 5% CO₂ and 10% FBS. 0.25 mg/mL Hoechst 33342 was used for tracking cells.
**b**, **c**, **d** Quantification of migration parameters. MSD: mean square displacement. Error bars in (**c**) and (**d**) represent 2×SEM (95% CI). **e** Representative tracks of chemotaxis in μ-Slide chemotaxis chamber (ibidi). Cells were allowed to migrate at 37 °C with 5% CO₂ in the presence of 1–20% FBS gradient. 0.25 mg/mL Hoechst 33342 was used for tracking cells. **f**, **g**, **i**, **j** Quantification of migration parameters. Error bars in (**g** and **i**) represent 2×SEM (95% CI). **h** Schematic of displacement: d,

distance: D, and forward displacement: y. Persistence ratio was defined as d/D, while Forward migration index was defined as y/D. Box whisker plots represent median, 1st, 3rd quartiles and 1.5×inter-quartile range. **b**, **f**, **j** Data were obtained from 3 or more independent experiments. In each experiment, n > 100 cells were analyzed for each cell lines. Steel-Dwass test (two sided) was performed and p values against DKO/p85β-wt were indicated. P values indicated as **** in (**b**) and (**f**) were <1.0×10⁻¹². In (**j**), p values of Steel-Dwass test were 6.0×10⁻⁴ for wt-DKO, 0.041 for wt-DKO/p85α-wt, and 9.0 × 10⁻⁴ for wt-DKO/p85β-motifGS, respectively, while the other pairs were not significant.

many of the membrane anchors that led to the p85-mediated endocytosis (Supplementary Fig. 4d, e) colocalize with ordered lipid domains. Both properties are known to form unique molecular organizations such as liquid droplets and lipid rafts. It is thus intriguing to speculate that it is this unique lipid-protein interaction that results in the biomolecular organization prerequisite for the p85-mediated endocytosis.

PI3K activity at focal adhesions is a major driver of mesenchymal cell migration. Earlier works showed that mesenchymal cells initiate protrusion with filopodia extension from nascent adhesions and that a positive feedback loop consisting of PI3K and actin dilates these adhesion-associated protrusions to develop mature lamellipodia[23,24]. Given that p85β has greater affinity to focal adhesion than p85α[64], p85β is assumed to play a dominant role in cell migration. We determined that AP2 binding of p85β negatively regulates its focal adhesion residence. As extension/retraction of membrane protrusions and their lifetime are all proportional to the PI3K activity[78], this AP2-mediated sequestration of p85β could act as a brake for migrating cells. Indeed, our data indicated that speed and persistency of cell migration correlate with the extent of p85β localization at focal adhesions. Furthermore, the AP2-mediated sequestration could fulfill a condition for long-sought negative feedback regulation of the PI(3,4,5)P$_3$ excitability[24]. Further exploration of molecular mechanisms underlying the observed p85β dissociation from focal adhesion should help reveal the understudied negative feedback regulation.

Of great interest, iSH2-mediated endocytosis is specific to the β isoform and not observed with the α or γ isoforms. Their opposing effects are reported elsewhere. For instance, p85α and p85β act as a tumor-suppressor and an oncogene, respectively[64,79–82]. Such a difference may have something to do with the endosomal PI3K signaling driven by p85β, but not by p85α. Recent studies revealed a role of endosomal PI(3,4,5)P$_3$ in Akt signaling[10,83]. In addition, the AP2 binding motif region coincides with the hinge region that determines the oncogenicity of p85β[82]. Thus, iSH2-mediated endocytosis possibly contributes to hyperactivate endosomal PI3K-Akt signaling. Furthermore, focal adhesion localization of p85β, which we found is regulated by AP2 motifs, was recently reported to be associated with kidney cancer[84]. T cell regulation may also be a target of p85β endocytosis. It was shown that T cell coreceptor CD28 preferentially binds to the p85β isoform[85], and that a PI3K-dependent endocytic process determines the CD28 pathway activity[86]. It is therefore tempting to speculate that iSH2-mediated endocytosis associates with the enigmatic difference in immune phenotypes between p85α and p85β knockout mice[1,11,87,88]. Accordingly, the impact of p85β-mediated endocytosis on physiological functions, as well as the molecular mechanisms leading to the difference between α and β, are fundamental to comprehensive understanding of the multi-faceted PI3K molecule in both normal and cancer cells.

We note several limitations of our study. First, how the p85 isoform specificity in endocytosis induction arises remains unknown. Given both α and β isoforms interact with AP2 (Supplementary Fig. 2), α and β isoforms may have distinct binding kinetics against AP2 (such as off rates). Therefore, kinetic analysis of AP2-p85 interaction and its influence on in vivo functions may be a key to better understand isoform specificity of PI3K regulation. Second, physiological relevance of their isoform specific role in cell motility should be developed further by, for example, studying expression profiles of each isoform in cells and tissues from normal individuals and cancer patients, in addition to looking into their relationship with cell migratory properties. Lastly, while we discovered a link between p85β and AP2 and its role in focal adhesion localization and cell migration, our observation is primarily based on exogenous expression of tagged proteins. Thus, characterization of endogenous p85β including CRISPR mutagenesis of AP2 binding motifs in p85β and its effects on physiological functions at the level of tissues and organisms would be expected to enhance our knowledge.

## Methods

### Reagents and antibodies

Rapamycin was purchased from LCLab (R-5000), prepared as 100 μM stock solution in DMSO, and stored at −20 °C. Alexa Fluor 647 conjugated transferrin was purchased from Thermo Fisher Scientific (T23366), reconstituted with Milli-Q water to obtain 5 mg/mL stock solution in PBS, and stored at 4 °C. mCLING-ATTO 647N-labeled was purchased from Synaptic Systems (710 006AT1), reconstituted with Milli-Q water to obtain 50 μM stock solution in PBS, and stored at −80 °C. LY294002 was purchased from Selleck Chemicals (S1105), prepared as 50 mM stock solution in DMSO, and stored at −20 °C. Fibronectin was purchased from Sigma-Aldrich (F4759), reconstituted with Milli-Q water to obtain 1 mg/mL stock solution, and stored at −20 °C. Once frozen fibronectin was thawed, the remainder was kept at 4 °C. PDGF-BB was purchased from Sigma-Aldrich (P3201), reconstituted with 4 mM HCl containing 0.1% BSA to obtain 50 μg/mL stock solution, and stored at −20 °C. FAK inhibitor PF-573228 was purchased from Selleck Chemicals (S2013), prepared as 20 mM stock in DMSO, and stored at −20 °C. Hoechst 33342 (10 mg/mL solution in water) was purchased from Thermo Fisher Scientific (H3570) and stored at 4 °C. Vinculin antibody (MAB3574-25UG) was purchased from Sigma-Aldrich. Akt (9272 S), phospho-Akt (T308) (13038 S), FAK (13009 S), and phospho-FAK (Y397) (8556 S) antibodies were purchased from Cell Signaling Technology. GAPDH antibody (sc-32233) was purchased from Santa Cruz Biotechnology. Mouse Adaptin α antibody was purchased from BD Biosciences (BD610501). Alexa Fluor 488-conjugated anti-Rabbit IgG (A-21206), Alexa Fluor 568-conjugated anti-Mouse IgG (A11004), Alexa Fluor 647-conjugated anti-Mouse IgG (A-31571), and Alexa Fluor 647-conjugated transferrin (T23366) were purchased from Thermo Fisher Scientific. Anti-IgG (H + L chain) (Mouse) pAb-HRP is purchased from MBL (MBL 330).

### Plasmids

The sequences of Lyn[89], KRasCAAX[90], EYFP-FKBP[91], EYFP-FKBP-iSH2β(mouse)[91], and PH(Akt)[45] have been reported elsewhere and their plasmids are summarized in Supplementary Data 2. The other plasma membrane anchors were constructed based on Lyn-ECFP-FRB or FRB-ECFP-KRasCAAX by replacing membrane anchor sequences with synthesized oligo DNA. ORF sequences of the plasma membrane anchor series are summarized in Supplementary Data 1[92–94]. Of note, LAT-ECFP-FRB was tagged with Kir2.1 signal (RAQLLKSRITSEGEYIPLDQI-DINVGFDSG) and ER export signal (NANSFCYENEVALTSK) to maximize plasma membrane localization[95]. EYFP-FKBP-iSH2β(mouse)-DN was constructed by deleting M470−R504 by inverse PCR with the primer set (fwd: 5′-GCTGCAGCGAGAGGGAAATGAGAAG-3′, rev: 5′-CCTCTCGCTGCAGCTCCTGGGAGGT-3′). iSH2β(mouse)-Δmotif was PCR-amplified with template plasmid EYFP-FKBP-iSH2β(mouse) and the primer set (fwd:5′-GCTGGTGGTCCTCGAGCATCCAAGTACCAA-CAAGACCAGG-3′, rev: 5′-AATTGAATTCTCAAGTCTCGTTCTTGATTC CCAG-3′) and inserted between XhoI and EcoRI sites by restriction digestion and T4 ligation. iSH2β(mouse)-motif-3×SAGG was similarly PCR-amplified with the template plasmid EYFP-FKBP-iSH2β(mouse) and with the primer set (fwd:5′-GCTGGTGGTCCTCGAGCATCCAAG-TACCAACAAGACCAGG-3′, rev: AATTGAATTCTCACGTGCGCTCCTC GTGGTGGGGGAGGCCTCCGGCAGACCCGCCTGCGGAGCCTCCAGCG CTAGTCTCGTTCTTGATTCCCAG) and inserted between XhoI and EcoRI sites. Alanine mutants of motif sequences were created by inverse PCR with corresponding primer sets.

mCherry-Rab5(C. lupus) and LAMP1(human)-mRFP were kind gifts from Dr. Gerald R.V. Hammond. mCherry-KDEL, mCherry-Dyn(WT), and mCherry-Dyn(K44A) were constructed by replacing the fluorescent protein portion of YFP-KDEL[96], YFP-Dyn(WT), and YFP-

Dyn(K44A)[90] with restriction digestion and T4 ligation. AP180(rat)-mCherry was a kind gift from Dr. Justin W. Taraska. To make the truncated version AP180C-mCherry, AP180 (530−918 aa) was PCR-amplified with the primer set (fwd: 5′-CTTCGAATTCTGGCCAC-CATGGCTGCCGCCACCACC-3′, rev: 5′-CGGTGGATCCCCCAAGAAAT CCTTGATGTTAAGATCCGCTAATGG-3′) and inserted into EcoRI and BamHI sites of pmCherry-N1 (Clontech) by restriction digestion and T4 ligation. AP2μ2(rat)-mCherry was obtained from Addgene (#27672). AKT-KTR-mRuby2 was designed based on a previous work[97]. In short, mouse FOXO1 from 1 to 380 AAs was fused to mRuby2. To eliminate DNA binding of FOXO1, S209A, H210A, S215A, and K219Q mutations were introduced.

The plasmids of mouse p85α, human p85β, and human p55γ were obtained from Addgene (#1407, #70458, # 70459). The plasmid of human p85α was obtained from DNASU. To construct EYFP-FKBP-iSH2α(mouse), EYFP-FKBP-iSH2α(human), EYFP-FKBP-iSH2β(human), and EYFP-FKBP-iSH2γ(human), each iSH2 region was PCR-amplified with the template of corresponding p85 or p55 plasmid and the primer sets (mouse-α-fwd: 5′-GGTCCTCGAGCATCCAAATACCAGCAGGATCA AGTTG-3′, mouse-α-rev: 5′-TGCAGAATTCTCACGTCTTCTCGTCATGG TGGG-3′, human-α-fwd: 5′-ATATCTCGAGCATCCAAATACCAACAGGAT CAAGTTG-3′, human-α-rev: 5′-ATATGAATTCTCACCATGTCTTCTCAT CATGATGGGG-3′, human-β-fwd: 5′-GCTGGTGGTCCTCGAGCTTCCA AATACCAGCAGGACCAG-3′, human-β-rev: 5′-GTCGACTGCAGAATTCT CAAGTGCGTTCCTCGTGG-3′, human-γ-fwd: 5′-GCTGGTGGTCCTCGA GCATCCAGATACCAACAGGATCAGTTG-3′, human-γ-rev: 5′-GTCGACT GCAGAATTCTCAGGTTTTCTCATCATAATGGGGC-3′) and inserted between XhoI and EcoRI sites of EYFP-FKBP by restriction digestion and T4 ligation or Gibson assembly.

EYFP-p85β(mouse) was constructed by inserting PCR-amplified p85β(mouse) (fwd: 5′-AGATCTCGAGCTAGTGCTGGTGGTAGTGCTGG TGGTAGTGCTGGTGGTAGTGCTGGTGGTAGTGCTGGTGGTATGGCAG GAGCCGAGG-3′, rev: 5′-TGCAGAATTCTCAGCGTGCTGCAGACG-3′) between XhoI and EcoRI with restriction digestion and T4 ligation. EYFP-p85β(mouse)-motifGS was constructed by inverse PCR and T4 ligation with the primer set pretreated with T4 polynucleotide kinase (fwd: 5′-GGCGGGTCTGCCGGAGGCCTCCCCCACCACGAGGA-3′, rev: 5′-TGCGGAGCCTCCAGCGCTAGTCTCGTTCTTGATTCCCAGC-3′). EYFP-p85β(mouse)-Δmotif, EYFP-p85β(mouse)-DN (deletion of M470−R504) were created by inverse PCR with the primer sets (motifGS-fwd: GGCGGGTCTGCCGGAGGCCTCCCCCACCACGAGGA, motifGS-rev: TGCGGAGCCTCCAGCGCTAGTCTCGTTCTTGATTCCCA GC, Δmotif-fwd: 5′-ACGAGACTCTCCCCCACCACGAGGAG-3′, Δmotif-rev: 5′-GGGGGAGAGTCTCGTTCTTGATTCC-3′, DN-fwd: 5′-GCTGCAG CGAGAGGGAAATGAGAAG-3′, DN-rev: 5′-CCTCTCGCTGCAGCTCCTG GGAGGT-3′). For lentivirus vector construction, EYFP-p85 and its mutants were subcloned into FUGW-puro lentivector (a kind gift from Reddy lab) by using AgeI and EcoRI sites. To construct FUGW-puro-Paxillin(human)-mCerulean3, human Paxillin sequence was PCR-amplified from the template pTriEx-mCherry-Paxillin (a kind gift from Yi Wu lab) with the primer set (fwd: 5′-ATCCCCGGGTACC GGGCTAGCGCCACCATGGACGACCTCGACGCCC-3′, rev: 5′-CATGGTG GCGACCGGTGAACCAGCACTACCACCAGCACTACCACCAGCACTACC ACCAGCACTGCAGAAGAGCTTGAGGAAGCAG-3′) and inserted into the AgeI site of FUGW-puro lentivector by Gibson assembly. The mCherry-PH-Akt lentiviral plasmid was generated based on a modified Puro-Cre vector (plasmid # 17408; Addgene, mCMV promoter, and no Cre encoding region).

To construct bacterial expression plasmids for GST-iSH2 variants, we first constructed GST-3C-TEV-6×His plasmid as follows. First, two oligo DNA sets (GST-3C-TEV-6xHis-1f: 5′-GGTTCCGCGTGGATCTGGT CTTGAGGTGCTCTTTCAGGGACCCGGCAGTCTCGAGGGTCTGTACAA GCGAATTCAG-3′, GST-3C-TEV-6xHis-1r: 5′- CTGAATTCGCTTGTACA GACCCTCGAGACTGCCGGGTCCCTGAAAGAGCACCTCAAGACCAGAT CCACGCGGAACC-3′, and GST-3C-TEV-6xHis-2f: 5′-TACAAGCGAATTC

AGGAGAACCTCTACTTTCAAAGCGATCATCATCATCATCATCACTAAA AATTCATCGTGACTG-3′, GST-3C-TEV-6xHis-2r: 5′-CAGTCACGATGAA TTTTTAGTGATGATGATGATGATGATCGCTTTGAAAGTAGAGGTTCTC CTGAATTCGCTTGTA-3′) were separately annealed. The annealed oligo DNAs were then inserted between BamHI and EcoRI sites of pGEX-2T, which was a kind gift from Dr. Miho Iijima, by Gibson assembly. Consequently, GST-3C-TEV-6×His plasmid has GST, 3 C protease cleavage sequence (LEVLFQGP), XhoI site, BsrGI site, EcoRI site, TEV protease cleavage sequence (ENLYFQS), 6×His-tag, and stop codon in this order. GST-3C-TEV-6×His was used to express and purify GST control. To construct GST-iSH2 variants, iSH2 regions were PCR-amplified with the primer set (fwd: 5′-CAGTCTCGAGTCCAAGTAC CAACAAGACCAGGTG-3′, rev: 5′-TCCTGAATTCCCGTGCGCTCCTCGT GG-3′) from the corresponding EYFP-FKBP-iSH2 plasmids. PCR fragments of iSH2 variants were inserted between XhoI and EcoRI sites of GST-3C-TEV-6×His plasmid by restriction digestion and T4 ligation.

## Protein purification

AP2 core was purified with slight modification from a previous report[98]. ECOS Competent E. coli BL21(DE3) (Nippon Gene 314-06533) was transformed with the two plasmids kindly shared by Dr. Gunther Hollopeter: pGH504 encoding mouse AP2 α2(trunk)-GST fusion and rat AP2 σ1 in chloramphenicol resistant vector and pGB21 encoding mouse b2(trunk)-6×His-tag fusion and mouse μ2 in ampicillin resistant vector. The bacteria were cultured overnight in LB broth supplemented with 100 μg/mL Carbenicillin and 25 μg/mL Chloramphenicol (LB/Carb/Cam). Pre-culture was inoculated to 2×YT media supplemented with 100 μg/mL Carbenicillin and 25 μg/mL Chloramphenicol (2×YT/Carb/Cam) and cultured at 37 °C until OD600 reached around 0.8–1.0. Protein expression was induced by adding 100 μM IPTG and allowed to proceed for 24 h at 18 °C. The cells were re-suspended with AP2 lysis buffer (1×PBS, 2.5 mM MgCl₂, 1 mM DTT, 1 mM PMSF) supplemented with 1 μg/mL NucA exonuclease (S. marcescens), 0.3 mg/mL Lysozyme, and 1× Protease Inhibitor V (Fujifilm Wako 168-26033) and lysed by sonication. The protein was bound to Glutathione agarose equilibrated with AP2 lysis buffer, washed with AP2 wash buffer (50 mM Tris-HCl (pH 7.5), 150 mM NaCl, 1 mM DTT, 1 mM PMSF), and eluted with 20 mM Glutathione in AP2 wash buffer. The sample was mixed with GST-tagged 3 C protease at 200:1 (w/w) ratio and dialyzed against AP2 dialysis buffer (25 mM Tris-HCl (pH 7.5), 150 mM NaCl, 1 mM DTT, 1 mM PMSF). Cleaved GST and GST-fused 3 C protease were removed by Glutathione agarose column and AP2 complex was further purified with Superdex 200 increase 10/300 column equilibrated with AP2 dialysis buffer. AP2 rich fractions were collected and concentrated by Vivaspin Turbo (Sartorius, 10 kDa cut off). The protein was stored at -20 °C in 25 mM Tris-HCl (pH 7.5), 150 mM NaCl, 30% glycerol, 1 mM DTT, 1 mM PMSF.

GST and GST-fused iSH2 proteins were purified as follows. BL21-CodonPlus (DE3)-RIL was transformed with GST protein-encoding plasmids. The bacteria were cultured overnight in LB broth supplemented with 100 μg/mL Ampicillin and 25 μg/mL Chloramphenicol (LB/Amp/Cam). Pre-culture was inoculated to LB media supplemented with LB/Amp/Cam and cultured at 37 °C until OD600 reached around 0.4–0.6. Protein expression was induced by adding 300 μM IPTG and allowed to proceed for 20 h at 18 °C. The cells were re-suspended with GST buffer (50 mM HEPES-NaOH (pH 7.5), 150 mM NaCl, 5 mM MgCl₂, 10% glycerol, 7 mM β-mercaptoethanol, 1 mM PMSF) supplemented with 1 μg/mL NucA exonuclease (S. marcescens) and cOmplete EDTA-free protease inhibitor cocktail and lysed by microfluidizer. The protein was bound to Glutathione agarose equilibrated with GST buffer, washed with GST buffer, and eluted with 20 mM Glutathione in GST buffer. The proteins were then applied to Ni-NTA agarose column equilibrated with Ni buffer and washed and eluted by stepwise increase of Imidazole in Ni buffer. Fractions enriched with GST-proteins (typically 50 to 500 mM Imidazole) were collected and dialyzed against

dialysis buffer (25 mM HEPES-NaOH (pH 7.5), 150 mM NaCl, 10% glycerol, 7 mM β-mercaptoethanol). The proteins were then concentrated by Vivaspin Turbo (Sartorius, 10 kDa cut off), aliquoted, snap frozen by liquid nitrogen, and stored at -80 °C.

### GST-pulldown assay

Ten microliters of Glutathione-agarose beads (Thermo Scientific 16101, 20 μL of 50% slurry) equilibrated with Buffer A (25 mM Tris-HCl (pH 7.5), 150 mM NaCl, 2 mM $MgCl_2$, 1 mM DTT) were mixed with 100 μL of 10 μM GST proteins and rotated at 4 °C for 30 min. The beads were then washed twice with 100 μL of Buffer A, mixed with 178 μL of 5.6 μM AP2 core, and rotated at 4 °C for 30 min. After two times wash with 50 μL buffer A, glutathione-bound proteins were eluted by rotation in 25 μL of buffer A supplemented with 20 mM Glutathione at 4 °C for 30 min. The samples were mixed with SDS-sample buffer, boiled at 95 °C for 5 min, and separated on polyacrylamide gel.

To quantitatively detect AP2, we performed western blot for the α subunit of AP2. After gel electrophoresis, proteins were transferred to methanol pre-treated PVDF membrane by using Criterion Blotter (BioRad, 1704070JA). The membrane was blocked by rocking in blocking buffer (3%BSA, 1×TBS) for 30–60 min at RT, stained with primary antibodies (anti-Adaptin α, BD Biosciences BD610501, ×100 dilution) by rocking in antibody buffer (3%BSA, 1×TBS, 0.1% Tween 20, 0.1% $NaN_3$) overnight at 4 °C, washed (5 min×3 times) with TBS-T, stained with secondary antibodies (Anti-IgG (H + L chain) (Mouse) pAb-HRP, MBL 330, ×5000 dilution) in buffer (3%BSA, 1×TBS, 0.1% Tween 20) for 1 h at rt, and washed again (5 min×3 times) with TBST. Chemiluminescence signals were detected by Chemi-Lumi One Ultra (Nacalai tesque, 11644-40) and ChemiDoc (Bio-Rad), and analyzed by Fiji software[99].

### Cell culture

HeLa (ATCC), Cos-7 (ATCC) and HEK293FT cells (a kind gift from the Andrew Ewald lab) were cultured in a DMEM (Corning, 10-013-CV) medium supplemented with 10% fetal bovine serum (Sigma-Aldrich, F6178). Wild type and p85 double knock out (DKO) mouse embryonic fibroblast (MEF) cells were kind gifts from the Brendan Manning lab and cultured in DMEM with 10% FBS.

### Generation of YFP-p85 rescued MEF cells

EYFP-p85 rescued cells were established by lentivirus transduction. Lentiviruses were produced by transfecting HEK293FT cells as follows. Five hundred microliters of Opti-MEM was mixed with 10 μg FUGW-puro-EYFP-p85, 7.5 μg Δ8.9, and 3.5 μg VSV-G plasmids. Another 500 μL of Opti-MEM was mixed with 63 μL of 1 mg/mL polyethylenimine. Two solutions were mixed and kept at room temperature for 20 min, then added to HEK293FT cells seeded one day before at $6 × 10^6$ cells/10 cm dish density. Two and three days after transfection, media were collected. The virus-containing media were mixed with 1/3 volume of 40% (w/v) PEG-8000, 1.2 M NaCl, 1×PBS (pH 7.0–7.2) and kept at 4 °C for more than 45 min. The viruses were precipitated by centrifugation (1500 × g for 45 min at 4 °C) and resuspended with PBS (200 μL for 10 cm dish cells). Aliquoted viruses were flash-frozen in liquid nitrogen and stored at -80 °C. To infect p85 DKO cells with the viruses, p85 DKO cells were seeded one day before infection at $4 × 10^4$ cells/well (6-well) density. On the day of infection, medium was replaced with fresh 500 μL of medium and virus suspension (10–100 μL depending on titer) and final 10 μg/mL polybrene were added. YFP positive cells were sorted by SH800S (SONY).

### Transient transfection

HeLa and Cos7 cells were transfected by lipofection with Xtreme-Gene9 (Sigma-Aldrich, 6365787001) in reverse transfection manner. Typically, 40 μL Opti-MEM, 1 μL XtremeGene9, and 0.5–1 μg of plasmid DNA were used for 2 wells (8-well, $75 × 10^3$ cells/well for Cos7 cells, $150–200 × 10^3$ cells/well for HeLa cells, $25–50 × 10^3$ cells/well for MEF cells) and incubated at 37 °C with 5% $CO_2$ and 95% humidity, for 16–24 h before imaging. 8-well chambers (154534) were poly-D-lysine (P6407-5MG) coated except for TIRF AP2 colocalization assay (strong adhesion stabilizes AP2 on the plasma membrane and interferes with the imaging). MEF cells were transfected either by lipofection with XtremeGene9 or by electroporation with Nucleofactor 2b. For electroporation, $2 × 10^6$ cells were resuspended with Nucleofactor kit T solution (+ supplement 1) and mix with 5 μg plasmid DNA. After zapping with T-20 protocol, 1 mL culture medium was quickly added to the samples and the cells were seeded on fibronectin coated 8-well chambers at the density of $25–50 × 10^3$ cells/well.

### Microscopes and imaging

Confocal imaging was performed on a spinning-disk confocal microscope. The microscope was based on an inverted Axiovert 200 microscope (Zeiss) and equipped with a spinning disk confocal unit (CSU10; Yokogawa) and triple-band dichroic mirror (Di01-T442/514/647, Semrock). Excitations of CFP, YFP, and mCherry were conducted with diode lasers and a semiconductor laser (COHERENT, OBIS 445 nm LX 75 mW, OBIS 514 nm LX 40 mW, OBIS 561 nm LS 50 mW), which were fiber-coupled (OZ optics) to the spinning disk unit. Images were taken with a Neo Fluor 40× objective (Zeiss) and a CCD camera (Orca ER, Hamamatsu Photonics) driven by MetaMorph or Micro-Manager 1.4 (Open Imaging). Images of live cell CID assay were typically taken every 1 min for 40 min. Epi imaging for the mCLING assay sample and ERKKTR live cell imaging was performed by an Eclipse Ti inverted fluorescence microscope (Nikon) equipped with a 60× oil-immersion objective lens and Zyla 4.2 plus sCMOS camera. TIRF imaging of focal adhesions was performed by an Eclipse Ti inverted fluorescence microscope (Nikon) equipped with a 100× oil-immersion TIRF objective lens and pco.edge sCMOS camera (PCO). Nikon microscopes were driven by NIS-Elements software (Nikon). Confocal imaging in Supplementary Fig. 7d, was performed on LSM800 laser scanning confocal system. 101.4 μm × 101.4 μm area was taken as 1048 pixel × 1048 pixel through a Zeiss Plan-Apochromat 63×/1.40 oil Ph3 lens with scan speed 4 (pixel dwell time 8.05 μsec). YFP was excited by 448 nm and 500–580 nm emission was detected with 80 μm pinhole. mCherry was excited by 561 nm laser and 570–650 nm emission was detected with 80 μm pinhole.

All live cell imaging was performed in imaging media containing DMEM (Corning, 17-205-CV) and 1×Glutamax (Thermo Fisher Scientific, 35050061) with temperature (37 °C), $CO_2$ (5%), and humidity control by a stage top incubator and a lens heater (Tokai Hit). For fixation, typically, cells were chilled on ice, washed 2 times with ice-cold PBS, fixed by fixation solution (4% paraformaldehyde and 0.15 % glutaraldehyde in PBS) for 10 min at room temperature, washed 2 times with ice-cold PBS, and stored at 4 °C in PBS.

Image processing and analysis were performed by Fiji software[99].

### Chemically-inducible co-recruitment assay

EYFP-FKBP was fused to iSH2 or indicated mutants, while FRB-CFP is tethered to the inner leaflet of plasma membrane using the CAAX-region of K-Ras. Upon rapamycin addition, FKBP binds to FRB which brings the bait (mVenus-FKBP-iSH2) and the prey capable of binding (AP2-mCherry or mCherry) to the plasma membrane. Recruitment of the bait and the prey to the plasma membrane were detected by TIRF microscopy as an increased fluorescence signal (Supplementary Fig. 6a–c). For quantification, after background subtraction, co-recruitment levels of prey were measured by increase in mCherry (prey) signal normalized to the intensity before rapamycin addition. Only cells showing at least 30% increase in mVenus (bait) intensity after Rapamycin addition were considered.

### Statistics and reproducibility

All the quantified data were obtained from 3 or more independent experiments except for Supplementary Fig. 10d. To statistically compare a pair of data, wilcox.test was used in R as Wilcoxon rank sum test. To statistically compare multiple data, pSDCFlig (Asymptotic option) of NSM3 library was used in R as Steel-Dwass test.

### Quantification of iSH2 puncta index

Following the method described in Supplementary Fig. 13 of a previous paper[100], we created 5×5 median-filtered images of YF-iSH2 images and divided the raw image by the filtered images. iSH2 puncta index was measured by quantifying standard deviation of cytosolic region of the divided YF-iSH2 images. To avoid including intensity fluctuation caused by plasma membrane, regions of interest were manually drawn. We used Cos7 cells for the analysis of iSH2 mutants and variants since the cell showed more homogenous background (e.g., in the case of negative control YF) than HeLa cells. In Supplementary Fig. 7d, where we used laser scanning confocal instead of spinning disk confocal, we used 10 × 10 median-filtered images to divide the raw images and calculated "skewness" of the intensity histogram.

### Western blot

$3.6 \times 10^5$ cells/well (6-well) were seeded -16 h before experiment. The cells were serum-starved for 5–6 h, stimulated as described in figure legends with 5% $CO_2$ at 37 °C. The reaction was stopped by directly replacing the culture media with 100 μL ice-cold RIPA buffer (Cell Signaling, 9806 S) supplemented with cOmplete protease inhibitor (1×, Roche, 11873580001), 1 mM PMSF, and phosphatase inhibitors (1× for each, Sigma P5726 and P0044). Since cooling on ice was not sufficient to stop dephosphorylation, it was critical to immediately replace the media with RIPA buffer. Soluble fraction was collected as supernatant after centrifugation ($14,000 \times g$ for 10 min at 4 °C) and the protein concentration was measured by Bradford assay. The samples were mixed with SDS-sample buffer, heated at 95 °C for 5 min, and separated on polyacrylamide gel. Proteins were transferred to methanol pre-treated PVDF membrane by using Criterion Blotter (BioRad, 1704070JA). The membrane was blocked by rocking in blocking buffer (3%BSA, 1×TBS) for 30–60 min at RT, stained with primary antibodies (Akt, pAkt, FAK, pFAK: ×1000 dilution; GAPDH: ×500 dilution) by rocking in antibody buffer (3%BSA, 1×TBS, 0.1% Tween 20, 0.1% NaN₃) overnight at 4 °C, washed (5 min×3 times) with TBS-T, stained with secondary antibodies (×2000 dilution) in antibody buffer for 1 h at rt, and washed again (5 min×3 times) with TBST. Fluorescent signals were detected by Typhoon or Pharos and analyzed by Fiji software[99].

### Transferrin uptake assay

Transferrin uptake assay was performed by following the previous literature. Briefly, MEF cells were serum starved in the imaging media containing DMEM (Corning, 17-205-CV) and 1×Glutamax (Thermo Fisher Scientific, 35050061) for more than 2 h and incubated with 250 μg/mL of Alexa Fluor 647-conjugated transferrin for indicated time. Cells were then chilled on ice, washed 3 times with PBS, washed 3 times with acid solution (0.2 M acetic acid, 0.5 M NaCl, pH 4.1), washed 3 times with PBS, fixed with 4% paraformaldehyde in PBS at room temperature for 10 min, and washed with 3 times with PBS. The amount of endocytosed transferrin was measured by quantifying cytosolic intensity of Alexa Fluor 647 in epi fluorescence images.

### Immunofluorescence

Immunofluorescence against vinculin was performed as follows. 25×10³ cells/well MEF cells were seeded on fibronectin-coated 8-well chambers and incubated overnight in DMEM supplemented with 10% FBS. Cells were then washed with PBS twice, fixed with 4% paraformaldehyde in PBS at room temperature for 15 min, washed again with PBS twice, permeabilized 0.1 % Triton X-100 in PBS at room temperature for 2.5 min, and blocked with blocking buffer (1% BSA in PBS) at room temperature for 30 min. Antibody against vinculin was used as ×500 dilution in the blocking buffer and the binding was performed at 4 °C overnight. The secondary antibody Alexa Fluor 568-conjugated anti-Mouse IgG was used as ×1000 dilution in the blocking buffer and the binding was performed at room temperature for 1 h. Each antibody binding steps were followed by 3 times of 5 min wash with TBST.

### Proliferation assay

For proliferation assay, 2.5–5×10⁴ cells were seeded on flasks, cultured in DMEM supplemented with 10% FBS for 50–72 h, and the final number of cells were counted. Doubling time was calculated by initial and final number of cells assuming the cell growth is exponential.

### Random migration assay

24-well plates were coated with 10 μg/mL fibronectin (5 μg/cm²) > 30 min at 37 °C. 1×10⁴ MEF cells were seeded and incubated in DMEM supplemented with 1% FBS for roughly 20 h. Cells were washed once with fresh DMEM supplemented with 1% FBS and the media were replaced with DMEM supplemented with 10% FBS and 0.25 μg/mL Hoechst 33342. Cells were left in a 37 °C and 5% $CO_2$ incubator for 2 h (Hoechst staining seemed to delay in the presence of fibronectin or collagen coating). Random migration was performed at 37 °C and with 5% $CO_2$ and humidity. Images were captured every 10 min for 16 h through DAPI channel and phase contrast and analyzed by TrackMate[101] plugin in Fiji software[99].

### Chemotaxis

Chemotaxis assay was performed on μ-slide chemotaxis chambers (ibidi, 80326) following the manufacturer's protocol. Briefly, 2.4 × 10⁶/mL WT MEF or 3.0×10⁶/mL p85 DKO and rescued MEF were seeded. After incubation at 37 °C with 5% $CO_2$ and 95% humidity for 2–3 h, the right reservoir was filled with imaging media supplemented with 1% FBS and 0.25 μg/mL Hoechst 33342 and the left reservoir was filled with imaging media supplemented with 20% FBS and 0.25 μg/mL Hoechst 33342. The chamber was further incubated for 2 h to allow the FBS gradient to be established. Chemotaxis was performed at 37 °C with 5% $CO_2$ and 95% humidity. Images were captured every 10 min for 16 h through DAPI channel and bright field and analyzed by TrackMate plugin[101] in Fiji software[99].

### Computational analysis of AP2–iSH2 interaction

To perform the computation, we download and installed the Local-ColabFold ver. 1.5.2[102–104] and NAMD ver. 3.0b4[105] to a Linux computer (Ubuntu 20.04.1) that is equipped with AMD Ryzen Threadripper PRO 3955WX 16-Cores, 128 GiB memory, and GeForce RTX3090 graphic boards. We used default parameters of LocalColabFold, except for employing alphafold2_multimer_v3 for model-type, and enabling template, amber, and use-gpu-relax flags. As parameters of NAMD simulation, we set the timestep to 2 fs, and initial NVT equilibration were performed for 500,000 steps before carrying the 50,000,000 steps of production process. For the other parameters of NAMD simulation, default values of CHARMM-GUI were employed. The simulation trajectory was saved each 1 ns and used to analyze the Root-mean-square deviation (RMSD) of all the protein atoms from the initial configuration. The resulting NAMD simulations were visualized and analyzed using Visual Molecular Dynamics (VMD) software[106].

### Reporting summary

Further information on research design is available in the Nature Portfolio Reporting Summary linked to this article.

## Data availability

The datasets generated during this study are available from the corresponding author upon request. Structure of PI3K and AP2 complex were obtained from the previously published PDB data, 2Y3A, 2XA7, and 2JKR. Source data are provided with this paper.

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

## Acknowledgements

We thank Brendan Manning for p85 double knockout cells; Andrew Ewald for HEK293FT cells; Sandra B. Gabelli for human p85α plasmid; Gerald R.V. Hammond for Rab5 and LAMP1 plasmids; Justin W. Taraska for AP180 plasmid; Yi Wu for Paxillin plasmid; Gunther Hollopeter for AP2 core plasmids; Sergi Regot for AKT-KTR plasmid. We also thank Yuta Nihongaki for technical assistance on lentivirus and FACS experiments. We appreciate Yoshihiro Adachi, Hiroshi Senoo, Miho Iijima, and Hiromi Sesaki for technical support on lentivirus and western blot experiments. We thank Kyoko Chiba, Yasunori Okamoto, and Shinichi Niwa for technical support on pulldown assay. We acknowledge generous support from the laboratory of Satoshi Murata and Shinichiro M. Nomura. FRIS CoRE (a shared research environment in Tohoku University) is also acknowledged. Our appreciation extends to Shigeki Watanabe, Yuuta Imoto, Atsuo Sasaki, Sho W. Suzuki, and Chuan-Hsiang Huang for insightful comments on the research project, and to our lab members including Hideki Nakamura, Allister Suarez and Helen D. Wu for fruitful discussions. We also thank Robert DeRose for manuscript proofreading and experimental support. This study was supported by the National Institutes for Health (R01GM123130, R01GM136858 and R35GM149329 to TI, T32GM007445 to CSG and AFP), the DoD DARPA (HR0011-16-C-0139 to TI), the PRESTO program of the Japan Science and Technology Agency (JPMJPR20KA to HTM), and MEXT/JSPS KAKENHI Grants (20H00618, 20H05971, 22K12255 to IK). HTM was supported by Postdoctoral Fellowships from the Japan Society for the Promotion of Science. ADR was supported by American Heart Association postdoctoral fellowship 23POST1027352.

## Author contributions

H.T.M. initiated the project. H.T.M., J.M., A.D.R. and T.I. designed the experiments. H.T.M., J.M., A.D.R., T.Y., A.F.P., and C.S.G. performed the experiments and analyzed the data under the supervision of T.I. N.T. performed experiments under the supervision of H.T.M. I.K. performed AlphaFold2 and MD simulations. H.T.M. wrote the manuscript in consultation with T.I., H.T.M., J.M., A.D.R., I.K., and T.I. edited the manuscript. All the authors contributed to the final version of the manuscript.

## Competing interests

The authors declare no competing interests.
