## [Peer Review File · Nature Communications]

Non-catalytic role of phosphoinositide 3-kinase in mesenchymal cell migration through non-canonical induction of p85 β /AP2-mediated endocytosisReviewers' Comments:

Reviewer #1:

Remarks to the Author:

This work described a novel observation that p85 β , one of the three regulatory subunits of PI3K, possesses a unique internal SH2 domain that interacts with the clathrin adaptor complex AP2 and triggers the internalization of p85 β but not other p85 isoforms. The authors showed that in fibroblasts lacking the α and β p85, re-expression of p85 β suppressed cell mobility but re-expression of AP2-binding deficient p85 β mutant increased cell mobility in a p110 dependent manner. Thus, the authors concluded that p85 β has a unique role in controlling cell migration by counteracting the migration-promoting function of PI3K, which is achieved by activating the AP-2 dependent internalization of p110 to decrease the availability of p110 on the plasma membrane. The experiments are well designed. Images and figures are high quality. The finding that p85 β by recruiting AP2 could play a dual role in regulating p110-promoted cell mobility is novel and attractive, and could potentially be significant to the better understanding of PI3K in cell migration and related diseases like cancer metastasis.

Major points:

1. The major concern comes from the physiological significance of this finding. It will be helpful if the authors show that the down-regulation of p85 β or mutations in the AP2-binding motif is positively associated with cancer aggressiveness according to TCGA data base. Or under certain disease or physiological condition (or special cell type), p85 α or p85 β is dominantly expressed to support higher or lower cell mobility, respectively.
2. Figure 4b and 4f both showed that the mobility of DKO cells is weaker than wt cells (expressing at least both α and β p85 or all three isoforms?). If re-expression of p85 β further suppressed the mobility of DKO cells, one would expect that the re-expression of p85 α recovered the mobility of DKO cells to a similar or even higher level comparing to wt cells. But Figure 4f and Extended Data Figure 10b showed this is not the case and re-expression of p85 α didn't increase the mobility of DKO cells. Why?
3. If the AP2-binding deficient p85 β mutants increase cell mobility in a p110-dependent manner, why the re-expression of the last three DN mutants in Figure 4b all suppressed cell mobility to a level lower than DKO and DKO/p85 β -wt? When the DN mutants lose their binding to p110, whether they still induce internalization of themselves should not matter to cell migration any more, thus they are expected to be non-functional to cell migration but not dominant negative as suggested by Figure 4b and Extended Data Figure 10b. Another interesting extended question is, how does PIP3 level or PIP3 dynamics change at the PM locations where p85 α , p85 β -wt, p85 β -motifGS, or p85 β - Δ motif expresses?

Minor points:

1. In Figure 3F, if there is significant difference between the DMSO treated DKO cells expressing WT or mutated p85 β should be statistically analyzed. One would expect that p85 β -motifGS and p85 β - Δ motif exhibit significantly higher value than p85 β -wt.
2. In Figure 4, there are two Panel d. Also, what does "MSD" stand for?
3. Figure legend of Extended Data Figure 9 is confusing. Were cells serum starved or FAK inhibitor treated?
4. Figure legend of Extended Data Figure 10 is too brief to understand easily. In Panel d, the left plot seems like response to 10% FBS but not PDGF.

Reviewer #2:

Remarks to the Author:

The manuscript by Matsubayashi et al., entitled "Non-catalytic role of phosphoinositide 3-kinase in mesenchymal cell migration through non-canonical induction of p85beta/AP-2-mediated endocytosis"

is well organized and well written manuscript, and scientifically important piece of the works in PI3K. The authors illustrated p85beta specific iSH2 domain function in clathrin mediated endocytosis in the context of focal adhesions and cell migration. We have raised the following questions that may help to strengthen the scientific contents and conclusion of the manuscript.

1. Authors have indicated the p85beta specific putative motifs, YxxΦ, di-leucine and acidic clusters. Since class IA PI3Ks are heterodimers, are these endocytic motifs unmasked by p110 catalytic subunits and accessible for AP2 interaction and clathrin-mediated endocytosis?
2. Authors are advised to change the sentences in discussion part "The iSH2 domain is characterized as a positive regulator of PI3K since it stabilizes and recruits the catalytic subunit p110 to the plasma membrane". iSH2 domain is not well defined to recruit the p110 catalytic subunit to the plasma membrane.
3. Have authors ruled out that mobile puncta visible in cytosol upon the expression of YF-iSH2 domain expression are the YF-iSH2 protein aggregates?
4. Authors have used the p85 α /p85 β double knockout mouse fibroblasts followed by re-expression of the p85beta constructs. The p85 α /p85 β constitute the majority of p85 adaptor subunits and there is tight regulation in the expression levels of the p85 adaptor subunits (p85 α , p85 β and p55gamma) and p110 catalytic subunits, often thought in 1:1 stoichiometry. Is there a reason for using p85 α /p85 β double knockout mouse fibroblasts instead of p85beta knockout mouse fibroblasts? Authors are advised to discuss this important point in the manuscript.
5. Authors indicate the role of p85beta and AP-2 interaction in the endocytosis of the focal adhesions. Authors have shown that re-expression of AP-2 motif deficient form of the p85beta mutant induces aberrant focal adhesions without impacting the focal adhesion signaling (e.g., FAK phosphorylation) and downstream signaling (Akt and ERK). However, the investigation of the role of p85beta and AP-2 interaction and endocytosis and its impact on downstream signaling seems less rigorously examined. It seems like the examination on FAK phosphorylation carried out in FBS containing and normally growing conditions, as result, not seeing any impact on the FAK phosphorylation as well as in Akt and Erk signaling.
6. Authors rely on Akt-PH domain based reporter assay to define the role in PI(3,4,5)P3 generation. The author may also examine Akt activation.
7. Ectopic expression/over expression is always a concern how can the authors address this issue?
8. Would a catalytic dead PI3K mutant be of use in these experiments confirm inhibitor results?

We thank the reviewers for the valuable and constructive feedback, each of which we took serious care to address. Of note, we now provide direct evidence of the interaction between AP2 and p85 β (Extended Data Figure 1b–j, 5c, d, and supplementary text). Please see below for our response to individual comments.

Reviewer #1 (Remarks to the Author):

This work described a novel observation that p85 β , one of the three regulatory subunits of PI3K, possesses a unique internal SH2 domain that interacts with the clathrin adaptor complex AP2 and triggers the internalization of p85 β but not other p85 isoforms. The authors showed that in fibroblasts lacking the α and β p85, re-expression of p85 β suppressed cell mobility but re-expression of AP2-binding deficient p85 β mutant increased cell mobility in a p110 dependent manner. Thus, the authors concluded that p85 β has a unique role in controlling cell migration by counteracting the migration-promoting function of PI3K, which is achieved by activating the AP-2 dependent internalization of p110 to decrease the availability of p110 on the plasma membrane. The experiments are well designed. Images and figures are high quality. The finding that p85 β by recruiting AP2 could play a dual role in regulating p110-promoted cell mobility is novel and attractive, and could potentially be significant to the better understanding of PI3K in cell migration and related diseases like cancer metastasis.

Major points:

1. The major concern comes from the physiological significance of this finding. It will be helpful if the authors show that the down-regulation of p85 β or mutations in the AP2-binding motif is positively associated with cancer aggressiveness according to TCGA data base. Or under certain disease or physiological condition (or special cell type), p85 α or p85 β is dominantly expressed to support higher or lower cell mobility, respectively.

While providing more supports on physiological relevance is indeed informative, we believe that our current dataset firmly revealing an unconventional role of a well-characterized kinase, PI3K, is sufficiently justified to be shared with the scientific community. Further exploration is underway and will be reported elsewhere in the near future. Nevertheless, we discuss this important point in the last paragraph of the Discussion section as a part of the “Limitations of our study”, as follows:

“Lastly, physiological relevance of their isoform specific role in cell motility should be developed further by, for example, studying expression profiles of each isoform in cells and tissues from normal individuals and cancer patients, in addition to looking into their relationship with cell migratory properties.”

2. Figure 4b and 4f both showed that the mobility of DKO cells is weaker than wt cells (expressing at least both α and β p85 or all three isoforms?). If re-expression of p85 β further suppressed the mobility of DKO cells, one would expect that the re-expression of p85 α recovered the mobility of DKO cells to a similar or even higher level comparing to wt cells. But Figure 4f and Extended Data Figure 10b showed this is not the case and re-expression of p85 α didn't increase the mobility of DKO cells. Why?

Since wt and DKO MEF cells were established from different mice (Ref # 62. Brachmann, S. M. et al.), there is no guarantee that rescuing both α and β genes in DKO cells fully recapitulate the properties of wt cells. Therefore, our intention was not to compare DKO cells with wt cells. Rather, we tried to compare phenotypes among DKO cells with different rescues.

3. If the AP2-binding deficient p85 β mutants increase cell mobility in a p110-dependent manner, why the re-expression of the last three DN mutants in Figure 4b all suppressed cell mobility to a level lower than DKO and DKO/p85 β -wt? When the DN mutants lose their binding to p110, whether they still induce internalization of themselves should not matter to cell migration any more, thus they are expected to be non-functional to cell migration but not dominant negative as suggested by Figure 4b and Extended Data Figure 10b. Another interesting extended question is, how does PIP₃ level or PIP₃ dynamics change at the PM locations where p85 α , p85 β -wt, p85 β -motifGS, or p85 β - Δ motif expresses?

p85 DN mutants can still bind to activated receptors albeit without accompanying the p110 catalytic subunit. This leads to failed PIP₃ production and poor cell migration, despite the presence of FBS (which contains receptor ligands such as growth factors). To clarify the action mechanism of DN mutants, we revised the description at Line 254 as follows:

“Both dominant negative mutation of p85 (DN), which lacks 470 to 504 aa residues necessary for p110 binding and thereby decouples receptors from PI3K signaling⁶⁵, and pharmacological inhibition of PI3K and FAK completely suppressed the migration.”

To quantify PIP₃ produced in DKO/p85 β -wt and DKO/p85 β -motifGS cells, we measured an extent of the PHAkt membrane-translocation upon PDGF stimulation, and found that there is no significant difference between these cells. We added the following description in the manuscript and the data is summarized in Extended Data Figure 10e.

Line 268

“When measured by PH(Akt) translocation to plasma membrane, wild-type p85 β -rescued and motifGS p85 β -rescued cells showed similar PI(3,4,5)P₃ response to PDGF stimulation within 30 min (Extended Data Figure 10e). Thus, PI3K signaling difference in longer time scale or subcellular activity may account for p85 β -AP2-mediated negative regulation of cell migration.”

Minor points:

1. In Figure 3F, if there is significant difference between the DMSO treated DKO cells expressing WT or mutated p85 β should be statistically analyzed. One would expect that p85 β -motifGS and p85 β - Δ motif exhibit significantly higher value than p85 β -wt.

Our motive was to show both wild type and mutant p85 reduce focal adhesion intensity “after FAK inhibitor treatment”. Therefore, the values were normalized by the intensity when the inhibitor was added (i.e., time 0). To quantitatively demonstrate the difference between the wild type and mutants, we newly quantified an adhesion/cytosol intensity ratio for each cell line followed by a statistical test (Extended Data Figure 9c, e).

2. In Figure 4, there are two Panel d. Also, what does "MSD" stand for?

We corrected the figure label, and clarified what MSD means in the legend (MSD: mean square displacement).

3. Figure legend of Extended Data Figure 9 is confusing. Were cells serum starved or FAK inhibitor treated?

To avoid confusion, we revised the corresponding part in the legend as follows:

“After serum starvation for one hour, the cells were imaged at 37°C with 5% CO₂ in the imaging media. At time 0, either 10 μM PF-573378 or DMSO was added.”

4. Figure legend of Extended Data Figure 10 is too brief to understand easily. In Panel d, the left plot seems like response to 10% FBS but not PDGF.

We revised the legend as follows:

“Extended Data Figure 10: Supplementary data of random migration assay. (a) Representative tracks of 2D random migration on fibronectin coated plates. (b) Mean speed of each condition of (a). PI3K inhibitor LY294002 and FAK inhibitor PF-573228 were added at 50 μM and 10 μM, respectively. Data set of (a, b) correspond with Fig. 4b-d. (c) Different data set of random migration including DKO/p85α-wt. MSD: mean square displacement. (d) Different data set of random migration stimulated by 10%FBS, 50 ng/mL PDGF, or 1 ng/mL PDGF. (c, d) Steel-Dwass test was performed and p-values against DKO/p85β-wt were indicated. P-values: ****: < 0.0001. ***: < 0.001. n.s.: not significant.”

Reviewer #2 (Remarks to the Author):

The manuscript by Matsubayashi et al., entitled “Non-catalytic role of phosphoinositide 3-kinase in mesenchymal cell migration through non-canonical induction of p85beta/AP-2-mediated endocytosis” is well organized and well written manuscript, and scientifically important piece of the works in PI3K. The authors illustrated p85beta specific iSH2 domain function in clathrin mediated endocytosis in the context of focal adhesions and cell migration. We have raised the following questions that may help to strengthen the scientific contents and conclusion of the manuscript.

1. Authors have indicated the p85beta specific putative motifs, YxxΦ, di-leucine and acidic clusters. Since class IA PI3Ks are heterodimers, are these endocytic motifs unmasked by p110 catalytic subunits and accessible for AP2 interaction and clathrin-mediated endocytosis?

To test this, we repeated the iSH2-induced endocytosis assay now with the p110α overexpression (Extended Data Figure 7d, e). Interestingly, overexpressed p110α, but not its kinase-dead mutant (D915N), suppressed the iSH2-mediated endocytosis. We speculate that unmasking of these AP2 binding motifs by p110 may require an additional factor, such as phospho-tyrosine (PMID 21362552, ref #16). Alternatively, AP2-mediated endocytosis may be driven by p85 molecules that are not bound to p110, in a manner similar to other p85 functions such as the regulation of unfolded protein response factor XBP-1 (PMID 20348923, PMID 20348926, ref 25, 26). We describe this result as follows:

Line 167

“We further tested the role of the p110 catalytic subunit in iSH2-mediated endocytosis. Expression of kinase-dead p110α (D915N) did not affect iSH2-mediated endocytosis compared to the mCherry control (Extended Data Figure 7d, e), consistent with a dispensable role of the PI3K catalytic activity in iSH2-mediated endocytosis. In contrast, expression of wild type p110α suppressed iSH2-mediated endocytosis, which appeared to be dependent on the expression level of p110 (Extended Data Figure 7d, e). While the mechanism by which wild type p110 suppresses iSH2-mediated endocytosis remains unknown, excessive PI(3,4,5)P₃ production due to overexpressed p110 may sequester AP2 and/or clathrin molecules from iSH2. Nevertheless, these data support the notion that iSH2-mediated endocytosis is independent of the p110 kinase activity.”

2. Authors are advised to change the sentences in discussion part “The iSH2 domain is characterized as a positive regulator of PI3K since it stabilizes and recruits the catalytic subunit p110 to the plasma membrane”. iSH2 domain is not well defined to recruit the p110 catalytic subunit to the plasma membrane.

Unless we misunderstand something here, we believe that iSH2 is responsible for bringing p110 to the active receptors at the plasma membrane. Besides revising an ambiguous expression, we added two more references (PMID 8313896, 8139567) to support this statement (below).

Line 290

“The iSH2 domain is characterized as a positive regulator of PI3K since iSH2 domain is necessary to engage the catalytic activity of p110 and link the catalytic subunit to receptor-binding p85^{63,75,76}”

3. Have authors ruled out that mobile puncta visible in cytosol upon the expression of YF-iSH2 domain expression are the YF-iSH2 protein aggregates?

Based on a series of our original results, we concluded that iSH2 puncta are membrane bound, instead of simple aggregates of YF-iSH2 proteins. This conclusion is supported, for example, by the iSH2 puncta colocalizing with a lipid membrane staining dye, mCLING (Extended Data Figure 2), and endomembrane markers (Fig 1c). In addition, the iSH2-mediated endocytosis was suppressed by dominant negative inhibition of clathrin and dynamin pathway (Fig. 1f, g) and introduction of mutations to iSH2 (Fig. 1d and Extended Data Figure 5), strongly supporting that these puncta are the result of endocytosis.

Nevertheless, there is a possibility that iSH2 forms aggregates at endocytic vesicles. Thus, we decided to test that possibility by using 1,6-hexanediol, a chemical that interferes with weak hydrophobic interactions and disperse protein condensates *in vitro* and in cells (Kroschwald et al., *Matters*, 2017, PMID: 26238190). With a stress granule marker Dcp1a (Kroschwald et al., *eLife*, 2015, PMID: 26238190), we confirmed 10% (w/v) 1,6-HD is sufficient to disperse protein condensates. At this concentration, 1,6-HD significantly reduced the index of iSH2 endocytosis (see below, the data is shared for the review purpose only). This data suggests that protein clustering or LLPS is involved in the generation step and/or maturation process of endocytic vesicles.

We decided not to include this data in the manuscript because of its inconclusive nature. More specifically, protein clustering or liquid-liquid phase separation (LLPS) is known to contribute to general endocytosis (PMID: 34887356, PMID: 33820972, PMID: 33688043). Thus, it is unclear if the observation with 1,6-HD is due to un-clustering of iSH2 aggregates, if any, or due to inhibition of the endocytic process.

Figure legend

Left: Confocal images of YFP-Dcp1a droplets with 0–10% 1,6-hexanediol. Cos7 cells were transfected with YFP-Dcp1a. The extent of droplet formation was quantified by skewness of YFP intensity histogram. **Right:** Confocal images of iSH2 puncta with or without 10% 1,6-hexanediol. Cos7 cells were transfected with Lyn-CR, YF-iSH2, and mCherry-PH(Akt). The extent of iSH2 vesicle formation was quantified by skewness of YFP intensity histogram. 0% 1,6-hexanediol, n=44 cells. 10% 1,6-hexanediol, n=37 cells.

4. Authors have used the p85 α /p85 β double knockout mouse fibroblasts followed by re-expression of the p85beta constructs. The p85 α /p85 β constitute the majority of p85 adaptor subunits and there is tight

regulation in the expression levels of the p85 adaptor subunits (p85 α , p85 β and p55gamma) and p110 catalytic subunits, often thought in 1:1 stoichiometry. Is there a reason for using p85 α /p85 β double knockout mouse fibroblasts instead of p85beta knockout mouse fibroblasts? Authors are advised to discuss this important point in the manuscript.

As we were interested in the difference between the two subtypes, p85 α and p85 β , we chose DKO cells as a common platform to test a role of each gene by rescuing it one by one. This intention could have been clearer if we have rescued the DKO cells with both genes.

To clarify the reasoning of the use of p85 DKO, we added the following sentences.

Line 204

“DKO allowed us to study mutations in AP2 binding motifs of p85 β as well as isoform difference between p85 α and p85 β in p85 knockout background.”

5. Authors indicate the role of p85beta and AP-2 interaction in the endocytosis of the focal adhesions. Authors have shown that re-expression of AP-2 motif deficient form of the p85beta mutant induces aberrant focal adhesions without impacting the focal adhesion signaling (e.g., FAK phosphorylation) and downstream signaling (Akt and ERK). However, the investigation of the role of p85beta and AP-2 interaction and endocytosis and its impact on downstream signaling seems less rigorously examined. It seems like the examination on FAK phosphorylation carried out in FBS containing and normally growing conditions, as result, not seeing any impact on the FAK phosphorylation as well as in Akt and Erk signaling.

Our motive to employ FBS was to assure the same condition as the one for the migration assay shown in Figure 4. Here, we intended to examine if faster-migrating cells with the p85 mutant rescue indicate any difference in FAK signaling. Nevertheless, we newly performed phosphor-FAK western blotting with PDGF, instead of FBS (Extended Data Figure 9a). As a result, there is no significant difference between wild type p85 and mutant p85, consistent with the FBS condition (Fig. 3e).

6. Authors rely on Akt-PH domain based reporter assay to define the role in PI(3,4,5)P3 generation. The author may also examine Akt activation.

As suggested, we monitored Akt activity using a molecular sensor known as Akt-KTR (kinase translocation reporter). When YF-iSH2, but not YF-iSH2-DN, was recruited to the plasma membrane, Akt-KTR translocated from nucleus to cytosol, indicating the Akt activation. This data is consistent with experiments with an Akt-PH domain, and also with our conclusion that iSH2-endocytosis is independent of PI3K catalytic activity. The data is summarized in Extended Data Figure 7a, b.

7. Ectopic expression/over expression is always a concern how can the authors address this issue?

It is indeed important to study endogenous proteins, either by monitoring or perturbing them. These could be achieved by performing antibody staining (for monitoring) or by introducing mutations into the AP2 binding motif (for perturbing).

Antibody staining: Since AP2-mediated endocytosis and migration regulation is β isoform specific, we attempted to detect endogenous p85 β isoform by using p85 β -specific antibody. We reached out to Prof. Ana C. Carrera, who recently developed p85 β -isoform specific polyclonal antibody K1123 (PMID: 25217619), only to learn that they could detect “overexpressed”, but not “endogenous”, p85 β . Nonetheless, we obtained the antibody and tried western blotting. However, p85 β was merely detectable due to a lot of non-specific bindings (see right, the data is shared for review purposes only).

Mutations into the endogenous locus: we are working toward this goal, and will report this elsewhere when ready.

In the revised manuscript, we now clarify this important point by adding “Limitations of our study” section in the last paragraph of the Discussion section as follows:

“While we discovered a previously uncharacterized link between p85 β and AP2 and its role in focal adhesion legalization and cell migration, our observation is primarily based on exogenous expression of tagged proteins. Thus, characterization of endogenous p85 β including CRISPR mutagenesis of AP2 binding motifs in p85 β and its effects on physiological functions at the level of tissues and organisms would be expected to enhance our knowledge.”

8. Would a catalytic dead PI3K mutant be of use in these experiments confirm inhibitor results?

As suggested, we repeated the iSH2-endocytosis assay with overexpression of a kinase-dead p110 mutant (D915N) (Extended Data Figure 7d, e), which resulted in production of endocytic vesicles similarly to the mCherry control condition. Thus, we confirmed that iSH2 endocytosis is independent of catalytic activity of PI3K.

Western blot using K1123,
which was raised by human
p85-beta 711–722 aa
(CRAPGGPPSAAR)

Reviewers' Comments:

Reviewer #1:

Remarks to the Author:

This reviewer's concerns and questions are satisfyingly addressed.

Reviewer #2:

Remarks to the Author:

The revised manuscript by Matsubayashi et al., entitled "Non-catalytic role of phosphoinositide 3-kinase in mesenchymal cell migration through non-canonical induction of p85beta/AP-2-mediated endocytosis" is well organized, well written manuscript, and has improved greatly. Also of the comments that we have raised have been fully addressed and this has greatly strengthen the scientific contents and conclusion of the manuscript. This manuscript should now be accepted. This is also consistent with recent reports from other group but is a completely novel direction.